# Frequency-dependent mobilization of heterogeneous pools of synaptic vesicles shapes presynaptic plasticity

Frédéric Doussau[1]*, Hartmut Schmidt[2], Kevin Dorgans[1], Antoine M Valera[1†], Bernard Poulain[1], Philippe Isope[1]

[1]Institut des Neurosciences Cellulaires et Intégratives, CNRS, Université de Strasbourg, Strasbourg, France; [2]Carl-Ludwig Institute for Physiology, University of Leipzig, Leipzig, Germany

**Abstract** The segregation of the readily releasable pool of synaptic vesicles (RRP) in sub-pools that are differentially poised for exocytosis shapes short-term plasticity. However, the frequency-dependent mobilization of these sub-pools is poorly understood. Using slice recordings and modeling of synaptic activity at cerebellar granule cell to Purkinje cell synapses of mice, we describe two sub-pools in the RRP that can be differentially recruited upon ultrafast changes in the stimulation frequency. We show that at low-frequency stimulations, a first sub-pool is gradually silenced, leading to full blockage of synaptic transmission. Conversely, a second pool of synaptic vesicles that cannot be released by a single stimulus is recruited within milliseconds by high-frequency stimulation and support an ultrafast recovery of neurotransmitter release after low-frequency depression. This frequency-dependent mobilization or silencing of sub-pools in the RRP in terminals of granule cells may play a role in the filtering of sensorimotor information in the cerebellum.

DOI: https://doi.org/10.7554/eLife.28935.001

*For correspondence: doussau@inci-cnrs.unistra.fr

Present address: †Department of Neuroscience, Physiology and Pharmacology, University College London, London, United Kingdom

Competing interests: The authors declare that no competing interests exist.

## Introduction

In neuronal networks, transfer of information largely relies on the ability of presynaptic terminals to convert changes in the firing rate of action potential (AP) into corresponding changes of neurotransmitter release. During trains of APs, the immediate tuning of synaptic efficacy is determined by the combination of multiple parameters including AP frequency, past firing activity, number of release-competent synaptic vesicles (SVs), also referred as the readily-releasable pool (RRP), vesicular release probability ($p_r$), and number of release sites ($N$). To date, the deciphering of cellular mechanisms underlying synaptic efficacy has been challenged by the contentious definition of the RRP (*Neher, 2015*). Depending on studies, the RRP either corresponds to a large population of docked SVs releasable by hyperosmotic stimulation or to SVs releasable by a single AP (*Moulder and Mennerick, 2005*; *Pan and Zucker, 2009*; *Neher, 2015*). Besides, there is increasing evidence that the RRP comprises heterogeneous populations of SVs that are differently poised for exocytosis and sequentially recruited for exocytosis. In the calyx of Held, for example, the RRP can be separated into a fast-releasing pool (FRP) and a slowly releasing one (SRP) (*Sakaba and Neher, 2001*; *Sakaba, 2006*; *Schneggenburger et al., 2012*). During a train of APs, SRP vesicles are first converted into FRP vesicles and then mature to a fully-releasable state (*Lee et al., 2012*, *2013*). In the cerebellar cortex, at synapses between granule cells (GCs) and molecular layer interneurons (MLIs), two subsets of the RRP have been described that are also mobilized by a two-step process, although the transition is faster than at the Calyx of Held (*Ishiyama et al., 2014*; *Miki et al., 2016*). Together, these studies suggest that short-term plasticity during high-frequency trains is shaped by the

differential mobilization of sub-pools of SVs in the RRP (*Pan and Zucker, 2009*; *Miki et al., 2016*). However, the mechanisms controlling these pools/steps upon a broad bandwidth of stimulation frequencies are still unknown.

Counter-intuitively, the size of the RRP does not determine the direction of presynaptic plasticity during high-frequency stimulation; at the calyx of Held with a very large RRP (700 to 5000 SVs; *Borst and Soria van Hoeve, 2012*) synaptic transmission quickly depresses, whereas at the GC to Purkinje cell (PC) synapse, synaptic transmission facilitates during tens of APs (*Kreitzer and Regehr, 2000*; *Valera et al., 2012*; *Schmidt et al., 2013*) despite a small RRP (4–8 SVs, *Xu-Friedman et al., 2001*) and a relatively high $p_r$ (*Schmidt et al., 2013*; *Sims and Hartell, 2005*; *Valera et al., 2012* - but see *Atluri and Regehr, 1996*). Using computer simulations combined with variance-mean analysis of postsynaptic responses, we previously showed that this striking facilitation is mediated by an increase in the number of release sites within milliseconds upon a $Ca^{2+}$-dependent process (*Valera et al., 2012*; *Brachtendorf et al., 2015*). Here, we investigated the cellular mechanisms underlying presynaptic short-term plasticity at GC-PC synapses not only during burst of activity but also upon fast changes in the stimulation frequency, as observed during in vivo recordings (*Chadderton et al., 2004*; *Jörntell and Ekerot, 2006*; *Rancz et al., 2007*; *Arenz et al., 2008*; *van Beugen et al., 2013*). We describe two subsets of SVs in the RRP with distinct properties: a fully-releasable pool that can be released by a single AP and a reluctant pool that is available for fusion within milliseconds at high stimulation frequencies. Moreover, we show that the fully-releasable pool can be specifically and almost completely silenced by low-frequency stimulation. The ultra-fast recruitment of reluctant SVs enables the large facilitation of glutamate release at high frequencies and the rapid recovery of neurotransmitter release following the near complete inactivation of fully-releasable SVs. We finally designed a model demonstrating how two pools of SVs can account for the low-frequency depression of glutamate release and its ultrafast recovery at high frequencies. These results support the idea that the distinct composition of the RRP in presynaptic terminals of GCs implements a dynamic filter of neuronal activity. This may explain how sensory inputs covering a wide range of firing rates are processed within the cerebellar cortex.

## Results

### Low-frequency depression at the granule cell-Purkinje cell synapse

In order to better understand the mechanisms underlying the strong facilitation of glutamate release during repetitive stimulations of the GC-PC synapse, we first measured synaptic transmission over a broad range of frequencies using transverse cerebellar slices prepared from young mice (P18 to P25). To focus on presynaptic mechanisms and to rule out postsynaptic contributions, we pharmacologically blocked the induction of postsynaptic long-term plasticity (LTP and LTD), NMDA-dependent plasticity (*Bidoret et al., 2009*; *Bouvier et al., 2016*), and endocannabinoid signaling (*Marcaggi and Attwell, 2005*; *Beierlein et al., 2007*). We stimulated parallel fibers (PFs) with trains of stimuli (50 pulses) elicited at different frequencies (2, 50 and 100 Hz) and at near-physiological temperature (34°C) (*Figure 1A*). As already reported in rodent cerebellar slices (*Kreitzer and Regehr, 2000*), a sustained facilitation of synaptic transmission was observed up to the 25th-30th stimulus at frequencies above 50 Hz, indicating that the number of SVs released increased with stimulation frequency. The paired-pulse facilitation at the first interval increased with frequency (paired-pulse facilitation $EPSC_2/EPSC_1$: 171.15% ± 10.03 at 50 Hz, $n = 9$, 215.96% ± 6.90 at 100 Hz, $n = 27$, *Figure 1A*). After 30 to 40 stimuli, a depression ($EPSC_n/EPSC_1 < 1$) was observed.

At many synapses, asynchronous release of quanta can be triggered by repeated stimulation (*Rudolph et al., 2011*; *Kaeser and Regehr, 2014*). Since asynchronous release can be detected by changes in the charge/amplitude ratio, we measured the charge and peak amplitude of EPSC during 50 Hz and 2 Hz stimulation. The perfect superimposition of normalized charges and amplitudes at any stimulus number during 50 Hz and 2 Hz trains suggested that our stimulation protocol did not evoke asynchronous release (*Figure 1A*). We then estimated the number of quanta released during stimulation. Values of quantal parameters described at unitary GC-PC synapses of mice (*Schmidt et al., 2013*) were used to estimate the number of quanta released at GC-PC synapse during stimulation at 2 Hz, 50 Hz and 100 Hz. The number of PFs stimulated in each experiment was estimated by dividing the mean value of EPSC amplitude at 0.033 Hz by the median EPSC amplitude

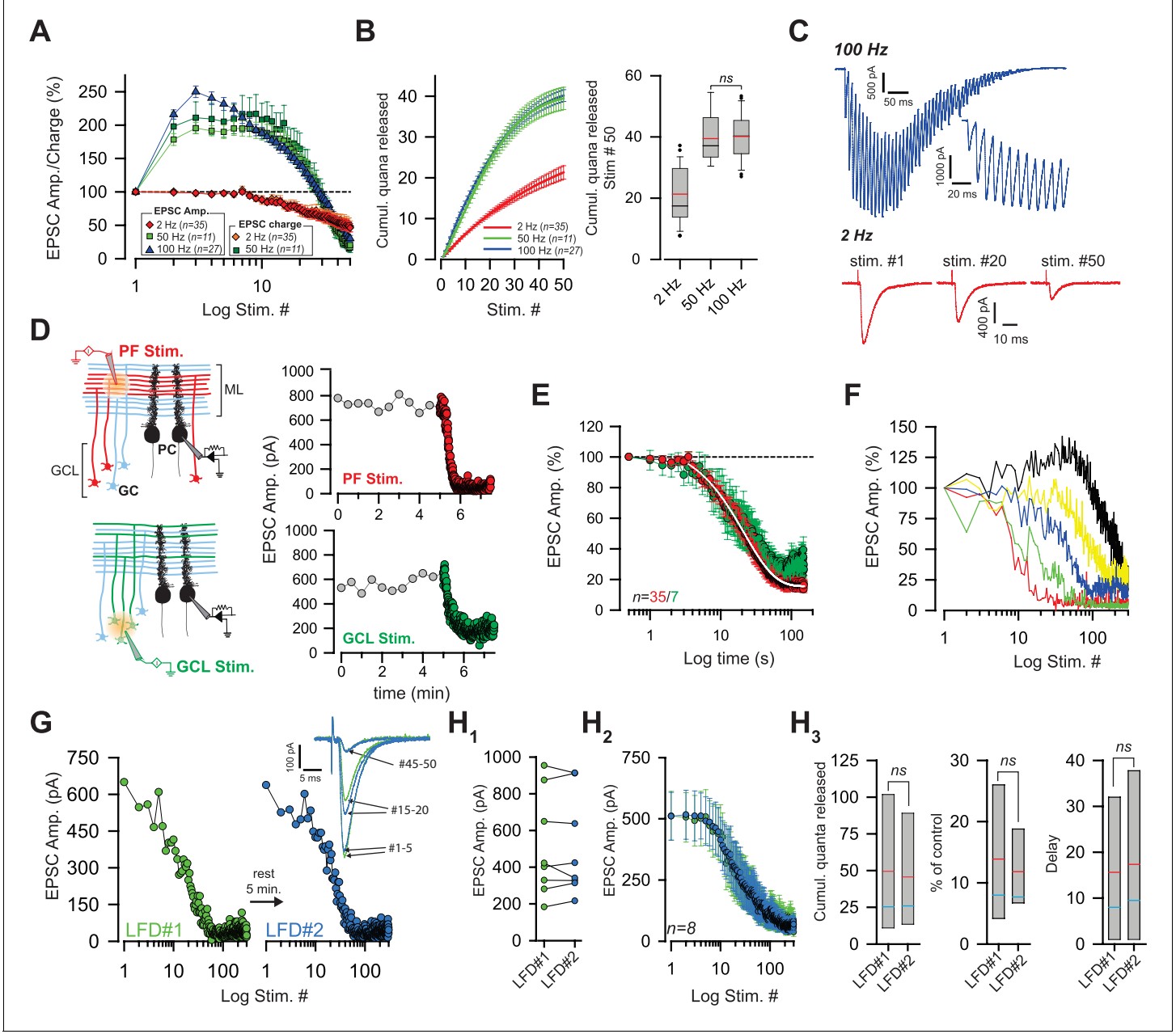

**Figure 1.** Low-frequency depression at GC-PC synapses. (**A**) Averaged EPSC amplitude and charge versus stimulus number during train of stimuli at 2, 50 and 100 Hz. At 50 Hz EPSC amplitude and EPSC charge are represented by light and dark green squares, respectively. At 2 Hz, EPSC amplitudes and EPSC charge are represented with red diamonds and orange diamonds, respectively. Note strong facilitation at 50 and 100 Hz, no facilitation at 2 Hz. (**B**) *Left* and *right panels* represent the number of quanta released during trains of 50 stimuli at 2, 50 and 100 Hz. *Left panel:* Cumulative number of quanta release versus stimulus number at 2 Hz (red line), 50 Hz (green line) and 100 Hz (blue line). Note that the superposition of values at 50 Hz and 100 Hz mask the values at 100 Hz. *Right panel:* Box-plots showing the corresponding cumulative number of quanta released at the 50th stimulus. No difference was observed between 50 and 100 Hz (*t*-test, p=0.82). Mean and median values are indicated in red and black, respectively. (**C**) Representative recording traces of EPSCs evoked at 100 Hz and 2 Hz. *Inset*: The first EPSCs observed during 100 Hz train. The stimulus artifacts have been subtracted for the 100 Hz train. (**D**) *left,* schematics showing the two modes of stimulation. Extracellular stimulations in the molecular layer (ML) (*upper panel*) activated beams of PFs (in red) whereas stimulations of the granule cell layer (GCL) (*lower panel*) led to sparse activations of PFs (in green) Non-stimulated GCs are represented in light blue. Stimulated areas are represented by concentric orange circles. *Right*, two examples of the time course of EPSC amplitudes following stimulation of PFs (*upper panel*) or of GC somata (*lower panel*) at 0.033 Hz (gray points) and 2 Hz (red or green points respectively). (**E**) Mean normalized EPSC amplitude during sustained 2 Hz stimulations of PFs or GC somata (red and green points respectively). Note the delay before the actual induction of LFD. The depression was fitted by a monoexponential function (tau = 21.7 s, white line). (**F**) Selected time course of LFD recorded in 5 PCs showing differences in the onset and the plateau of depression. (**G, H**) LFDs elicited in the same cell share the same

*Figure 1 continued on next page*

Figure 1 continued

profile of depression. (G) Time courses of two successive LFDs elicited in the same cell. The second LFD protocol (LFD#2, blue points) was performed after a resting period of 5 min (LFD#1, green points). *inset,* traces correspond to averaged EPSCs recorded during LFD#1 and LFD#2 (green and blue traces, respectively) at the indicated stimulus numbers. (H$_1$) EPSC amplitudes recorded at the first stimulus of LFD#1 (green points) and LFD#2 (blue points). The similar sizes of EPSC amplitudes in LFD#1 and LFD#2 indicates full recovery from depression during the resting period (paired *t*-test performed on EPSCs#1 of LFD#1 and LFD#2, p=0.94, *n* = 8). (H$_2$) Superimposition of mean EPSC amplitudes recorded during LFD#1 (green points) and LFD#2 (blue points). Same set of experiment as in H$_1$. (H$_3$) Box-plots showing the cumulative number of quanta released during LFD#1 and LFD#2 (number of quanta released during LFD: 49.5 quanta ± 18.8 quanta for LFD#1 versus 45.6 quanta ±, 14.5 quanta for LFD#2, paired *t*-test, p=0.47, *n* = 8), the plateau of LFD#1 and LFD#2 (mean percentage of initial response for LFD: 13.9 ± 5.2% for LFD#1 versus 11.8% ±, 3.2 for LFD#2, paired *t*-test, p=0.39, *n* = 8) and the delay before the onset of depression (15.6 stimuli ± 6.6 stimuli for LFD#1 versus 17.4 stimuli ± 7.2 for LFD#2. paired *t*-test, p=0.13, *n* = 8). Since none of these parameters were statistically different between the two conditions, LFD#1 and LFD#2 were considered identical. Blue and red lines indicate median and mean values respectively. Same set of experiments as in H$_1$, H$_2$.

DOI: https://doi.org/10.7554/eLife.28935.002

The following figure supplement is available for figure 1:

**Figure supplement 1.** LFD can be elicited by a broad range of frequencies.

DOI: https://doi.org/10.7554/eLife.28935.003

obtained at unitary GC-PC synapses (5.3 pA, including release failure, *Schmidt et al., 2013*). The number of quanta released at any synaptic terminal during each protocol was estimated by dividing the cumulative EPSC amplitude by the number of PFs stimulated and by the mean value of the quantal content obtained at unitary GC-PC synapse (8 pA, *Schmidt et al., 2013*). Our estimation does not take in account the ~10% of GC boutons that contain two active zones and make contact on two dendritic spines (*Xu-Friedman et al., 2001*). Therefore, the estimation of the number of quanta released at individual GC terminals have been underestimated through overestimations of the number of PFs recruited by external stimulation. Nevertheless, these errors should be constant in all the stimulation paradigms and we believe that the comparison between cumulative values of quanta released at individual GC terminals are relevant information. Cumulative EPSC amplitude plots demonstrate that the total number of quanta released is proportional to the stimulation frequency. However, although initial facilitation during the first two or three pulses is higher at 10 ms than at 20 ms intervals (*Valera et al., 2012*) no difference was observed in the maximal recruitment of releasable SVs between 50 Hz and 100 Hz (number of quanta released per bouton at 50 Hz = 39.47 ± 2.75, *n = 9* compared to 40.14 ± 7.50 at 100 Hz, *n = 27*, p=0.82, *t*-test, *Figure 1B*).

Conversely, at 2 Hz, the synaptic responses did not facilitate during the first stimuli (*Figure 1A,C*) and depressed rapidly after a mean delay of 7 stimuli ± 5 stimuli (*n* = 35) (*Figure 1D,E*). We named this rapid blockage of synaptic transmission 'low frequency depression' (LFD). Strikingly, the EPSC amplitudes decreased mono-exponentially in the majority of cells recorded (*n* = 32) with a mean time constant of 15.9 s ± 1.2 s. A lack of LFD was observed for ~12% of PCs recorded in the vermis (5 cells out of 39 cells). These cells were excluded from the statistical analysis.

APs are reliably initiated and transmitted along PFs at high frequencies and at physiologic temperature (35°C) (*Kreitzer and Regehr, 2000*; *Isope and Barbour, 2002*; *Baginskas et al., 2009*). Nevertheless, repetitive extracellular stimulations of PFs can decrease the excitability of fibers excitability due to K$^+$ accumulation in the extracellular space (*Kocsis et al., 1983*). To test, whether LFD was caused by decreased excitability of PFs, we placed the stimulation electrode in the granule cell layer (GCL) to stimulate clusters of GC somata rather than beams of PFs (*Figure 1D*). The lack of any significant change in the onset and the kinetics of depression after stimulating GC somata (*Figure 1D,E*) suggested that impaired PF excitability did not cause LFD.

While the onset and the plateau of LFD varied among PCs (*Figure 1F*), the depression could be reliably induced in a given PC as long as the recording of EPSCs could be maintained. In a series of 8 experiments, two LFD (LFD#1 and LFD#2) separated by a resting period of 5 to 10 min were successively induced. EPSC amplitudes fully recovered after the resting period following the end of LFD#1 protocol and the time courses of EPSCs amplitude of LFD#1 LFD#2 were identical (*Figure 1G,H*). Our findings reveal for a first time that a sustained stimulation at low frequency can almost completely silence the release apparatus of GC terminals.

In vivo recordings have shown that GCs fire spontaneously in a range of 0.4 to 10 Hz (*Chadderton et al., 2004*; *Jörntell and Ekerot, 2006*). Therefore, we applied successively at the

same synapse low-frequency stimulations in a range of 0.5 to 10 Hz (stimulations with random frequencies, *Figure 1—figure supplement 1A*) or at a fix low frequency (0.5 Hz, 2 Hz or 5 Hz, *Figure 1—figure supplement 1B,C*). No change in the properties of LFD were observed after a shift in the firing frequency (*Figure 1—figure supplement 1D,E*) indicating that presynaptic terminals of GCs can filter a broad range of low-frequency activities of GCs.

## Immediate recovery from LFD by high-frequency trains

High-frequency stimulations can induce rapid refilling of release sites and the recruitment of new ones (*Saviane and Silver, 2006*; *Hallermann et al., 2010*; *Valera et al., 2012*; *Chamberland et al., 2014*). Therefore, we tested whether an increase of stimulation frequency could fully restore release from presynaptic terminals after LFD induction. Alternatively, LFD may result from an activity-dependent blockage of presynaptic voltage-dependent calcium channels (*Xu and Wu, 2005*). In this latter case, accelerating the refilling of the RRP or increasing the number of release sites *N* should not restore glutamate release after LFD. LFD was induced by 300 stimuli at 2 Hz, and a train of 50 stimuli at 100 Hz was applied immediately after the last stimulus (*Figure 2A*). Strikingly, after a full blockage of synaptic transmission, the glutamate release capacity recovered to 70.7 ± 7.6% within 10 ms, reached a peak of amplitude after 5 to 8, and then declined progressively (*Figure 2B–D*). After the 7th stimuli, EPSC amplitudes evoked by the 100 Hz recovery train reached values that were not significantly different from values observed at the same stimulus number in a control train elicited before LFD induction (*Figure 2C,D*). This suggests that a full recovery from LFD was achieved within 70 ms.

We then studied how the frequency of stimulation in the train influenced the ability of depressed synapses to recover a full capacity of release. Accordingly, 50 Hz and 20 Hz trains were applied after LFD and compared to 50 Hz and 20 Hz trains applied in control conditions. Mean amplitudes of EPSCs recorded during train at 50 Hz applied in control condition were not statistically different from those recorded after LFD induction (*Figure 2E*) within approximately 50 ms (5 stimuli). With 20 Hz applied after LFD, the recovery was limited, and EPSC amplitudes were smaller than the ones recorded in control condition (*Figure 2F*).

## Recovery from LFD relies on the fast recruitment of reluctant SVs by a high-frequency train

In principle, the ultrafast recovery of synaptic transmission following LFD could be mediated by (1) a frequency-dependent ultrafast replenishment of the RRP and/or by (2) the recruitment of a reluctant pool of SVs that cannot be mobilized at low frequency as suggested in *Valera et al., 2012*. To test the latter mechanism, we studied whether the recovery from LFD by a 100 Hz train was affected by previous exhaustion of the reluctant pool. Our results suggested that the reluctant pool can be recruited by short bursts at high frequency (paired-pulse/triplet stimulation at 50 or 100 Hz, *Valera et al., 2012*). Therefore, we modified the stimulation paradigm for LFD and used 100 triplets of stimuli repeated at 2 Hz (LFD$_{triplet}$) instead of the 300 single pulses at 2 Hz (LFD). The total number of stimuli was the same in LFD$_{triplet}$ and LFD. The triplet frequency was changed from 10 Hz to 200 Hz during the experiment. Since LFD protocols can be repeated several times after a recovery time (*Figure 1G*), we systematically applied the LFD and LFD$_{triplet}$ protocols to each PC recorded. Strikingly, the level of recovery after LFD$_{triplet}$ was inversely proportional to the frequency of triplet stimulation (*Figure 3B,C*). For example, when triplets frequency was set at 200 Hz during LFD$_{triplet}$, synaptic transmission hardly recovered (maximal recovery 38.9%±7.19 $n$ = 6, compared to 196.7%± 8.7, after LFD, $n$ = 27, *Figure 3C*). This finding suggested that LFD$_{triplet}$ stimulation depleted or disabled the population of SVs that support the recovery from LFD. If SVs released only during LFD$_{triplet}$ (that is, at the second and third pulse of triplet stimulation) and during recovery by 100 Hz train belong to the same pool (the reluctant pool), then the number of quanta released during LFD and recovery train or during LFD$_{triplet}$ and recovery train should be the same. We next applied LFD and LFD$_{triplet}$ (triplet at 100 Hz) successively in the same cells followed by the application of a 100 Hz recovery train (*Figure 4A,B*). Cumulative EPSC amplitudes were determined, and the approximate number of quanta released for the whole protocol (LFD or LFD$_{triplet}$ +recovery train) was estimated (*Figure 4C*). The normalized cumulative EPSC amplitude was higher during LFD$_{triplet}$ than during LFD (6440.9 ± 945% for LFD compared to 9353.5 ± 1199.9% for LFD$_{triplet}$, p<0.001, paired *t* test,

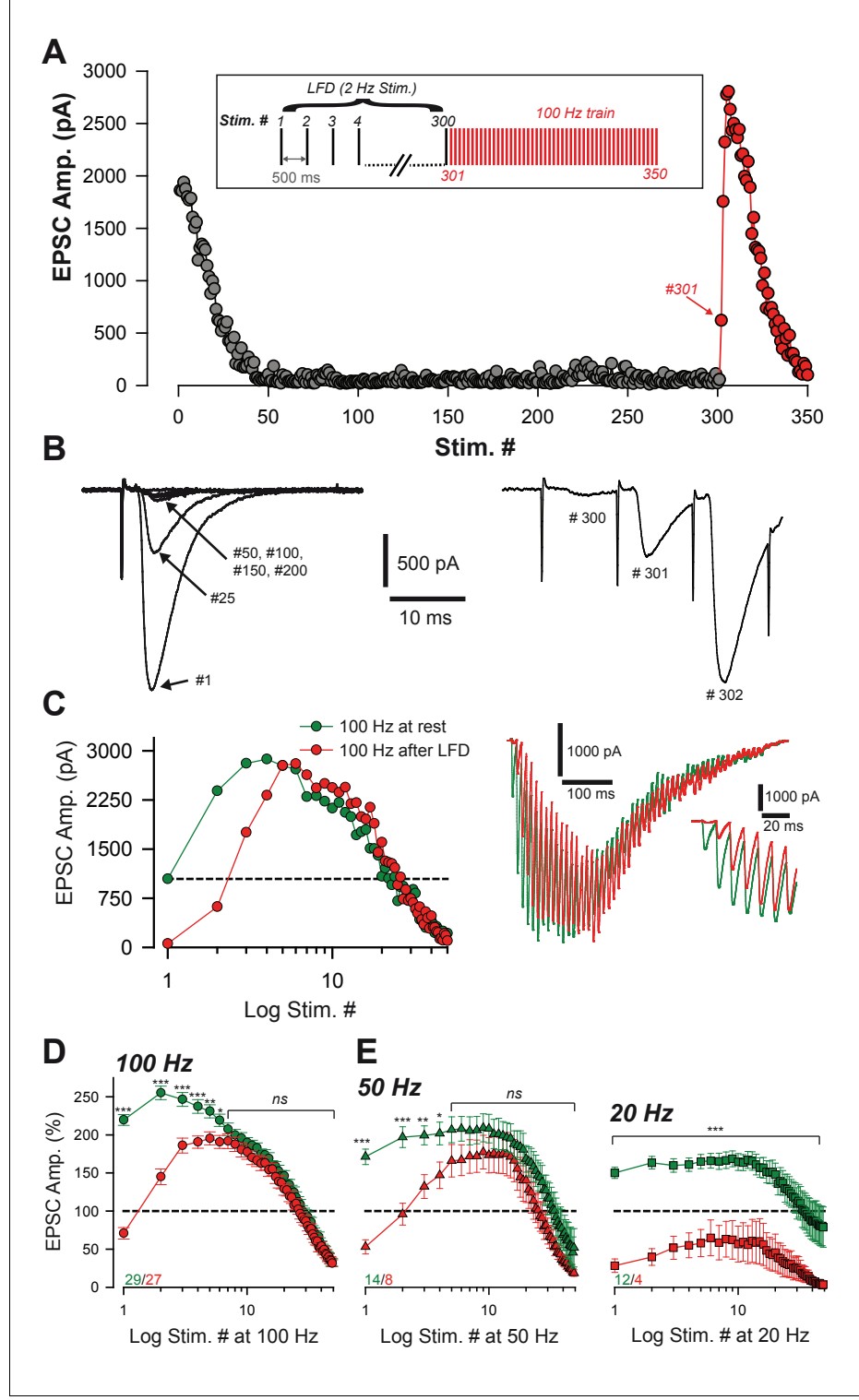

**Figure 2.** Ultrafast recovery from LFD by high-frequency trains. (**A**) Typical experiment illustrating the time course of EPSC amplitude during LFD (gray points, stimulation at 2 Hz) and the fast recovery from LFD via high-frequency trains (red points, stimulation 100 Hz). *Inset*: protocol of stimulation. Stimulus #1 corresponds to the beginning of the 2 Hz stimulation. (**B**) Recording traces of superimposed EPSCs recorded during 2 Hz stimulation (*left*) and during the following 100 Hz train (*right*) at the indicated stimulus number from (**A**). Note the ultrafast recovery from depression at 100 Hz (stimulus # 301). (**C**) *Left*, Representative EPSC amplitudes evoked by a 100 Hz train applied before (green), and 10 ms after 300 stimuli at 2 Hz. The dashed line corresponds to the baseline amplitude
*Figure 2 continued on next page*

*Figure 2 continued*
defined as mean amplitude at 0.033 Hz. Note the similar size of EPSCs after the fourth stimulus at 100 Hz. *Right,*
Corresponding traces recorded during these 100 Hz trains. *Inset*: The first EPSCs observed during train
application. (D) Mean values of normalized EPSC amplitude elicited by trains of stimulation at 100 Hz at baseline
(0.033 Hz, green) or after LFD induction (2 Hz, red). EPSC amplitudes were not significantly different after the
seventh stimuli (MWRST, p=0.181, *n* = 27). (E) Mean values of normalized EPSC amplitude elicited by trains of
stimulation at 50 Hz and 20 Hz at baseline (0.033 Hz, green) or after LFD induction (2 Hz, red). EPSC amplitudes
were not significantly different after the after the fifth stimuli for 50 Hz trains (*t*-test, p=0.137, *n* = 13). EPSC
amplitudes in 20 Hz trains elicited after LFDnever reached amplitudes of EPSCs of 20 Hz trains elicited at rest.
Numbers at the bottom of the graphs indicate the number of cells recorded under each condition.
DOI: https://doi.org/10.7554/eLife.28935.004

*n* = 18, *Figure 4C,D*). This suggests that reluctant SVs were recruited by the triplet stimulation. When the cumulative EPSC amplitude measured during the recovery train was added, no significant differences were observed between the two protocols (11696.3 ± 1163.6% for LFD and recovery train compared to 11379.0 ± 1349.6% for LFD$_{triplet}$ and recovery train, p=0.61, paired *t* test, *n* = 18, *Figure 4C,D*). This suggests that SVs recruited only during LFD$_{triplet}$ or only during recovery train belong to the same reluctant pool. Accordingly, the calculated number of quanta released during LFD was statistically lower than during LFD$_{triplet}$ (42.7 ± 6.2 quanta for LFD compared to 60.8 ± 8.0 quanta for LFD$_{triplet}$, p<0.01, paired t-test, n = 18, *Figure 4E*) and the total number of quanta released during LFD and recovery train or LFD$_{triplet}$ and recovery train were not significantly different (77.5 ± 7.7 quanta released for LFD and recovery train compared to 74.8 ± 8.9 quanta released for LFD$_{triplet}$ and recovery, p=0.51, paired *t*-test, *n* = 18, *Figure 4E*). Moreover, the number of quanta released during LFD and solely during recovery train (that is, at stimuli #301 to stimuli #350) were not significantly different (42.7 ± 6.2 quanta released for LFD compared to 32.1 ± 2.3 quanta released for recovery train, p=0.1, paired *t*-test, *n* = 18, *Figure 4E*). These results suggest that releasable SVs in GC boutons segregate in two distinct pools: the fully-releasable pool that can be released by a single AP and a reluctant pool that can only be recruited during bursts of APs reaching frequencies ≥ 20 Hz.

Physiologically, the duration of spontaneous activity of GCs preceding high-frequency bursts is variable. Therefore, the recruitment of the reluctant pool may occur regardless of the status of the fully-releasable pool. This latter pool may be fully available, partially silenced or fully silenced. Since strong paired-pulse facilitation indicates recruitment of the reluctant pool (*Valera et al., 2012*; *Miki et al., 2016*, this study), we studied how high-frequency paired-pulse stimulations randomly applied during the course of LFD influenced paired-pulse facilitation. Paired-pulse stimulation at high frequencies applied during LFD induced strikingly high PPR values (*Figure 4—figure supplement 1A,C*). These strikingly high PPR values were observed whatever was the percentage of inhibition and regardless of the timing of LFD (*Figure 4—figure supplement 1D*). These findings strongly suggest that the status of the fully-releasable pool does not influence the mobilization of the reluctant pool.

## A two-pool model accounts for basic experimental findings

Our data suggest a segregation of SVs into different pools at GC terminals, which is in line with several previous studies (*Valera et al., 2012*; *Ishiyama et al., 2014*; *Brachtendorf et al., 2015*; *Miki et al., 2016*). They further suggest that SVs can be recruited from a reluctant pool into a fully-releasable pool on a millisecond timescale. Several models can account for these findings (*Figure 5*), including a single-pool model with Ca$^{2+}$-independent replenishment, a sequential two-pool model with Ca$^{2+}$-dependent recruitment (*Millar et al., 2005*; *Sakaba, 2008*), and a parallel two-pool model with intrinsically different SVs in which both pools are restored independent of Ca$^{2+}$ (*Wölfel et al., 2007*; *Schneggenburger et al., 2012*). An important demand on such models is that it needs to integrate the observed high-frequency facilitation and LFD.

First, we analyzed a sequential two-pool model, where release is triggered via a Ca$^{2+}$-driven, five-site sensor from a fraction of releasable SVs (*Millar et al., 2005*; *Sakaba, 2008*) (*Figure 6A*, see Materials and methods). These SVs become replenished in two steps. The first step corresponds to a Ca$^{2+}$-dependent priming step (R$_0$→R$_1$) and the second one to a Ca$^{2+}$-independent filling step

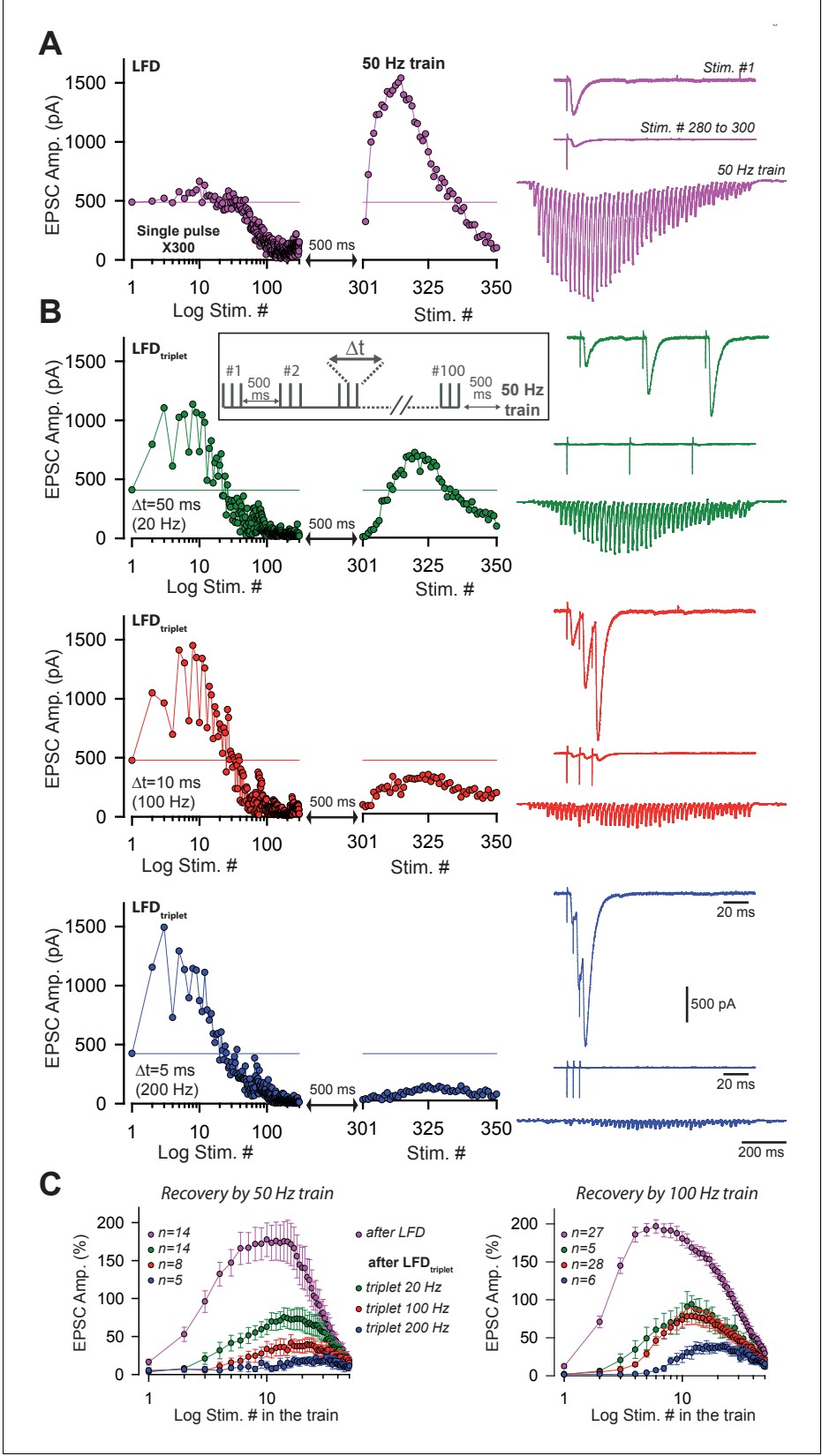

**Figure 3.** Short-term facilitation during triple-pulse stimulation at high frequency impedes recovery from LFD via high-frequency trains. (**A**) *Left panels* an example of EPSC amplitudes during LFD followed by a 50 Hz train of
*Figure 3 continued on next page*

*Figure 3 continued*

stimulation. *Right panel, upper trace*, EPSCs elicited by the first stimulation at 2 Hz, *middle traces*, averaged EPSCs recorded during the LFD plateau (stimuli #280 to #300), *bottom traces*, EPSC recorded during the 50 Hz train applied 500 ms after the 2 Hz stimulation (artifacts subtracted). (**B**) *left panels*, time courses of EPSC amplitudes induced by triples pulse at different frequencies (LFD$_{triplet}$, see inset for protocol) as indicated and by a subsequent 50 Hz train. *Right panel, upper trace*, EPSCs elicited by the first triplet at 2 Hz, *middle traces*, averaged EPSCs recorded during the LFD plateau (stimuli #80 to #100), *bottom traces*, EPSC recorded during the 50 Hz train applied 500 ms after the 2 Hz stimulation (artifacts subtracted). For *A* and *B*, all data and traces were obtained in the same PC, and LFDs or LFDs$_{triplet}$ were elicited after a resting period of 5 min after the end of each protocol. (**C**) Mean values of normalized EPSC amplitudes evoked by 50 Hz train (*left*) and 100 Hz train (*right*) applied 500 ms after LFD induction (300 single stimuli at 2 Hz or 100 triple pulses at 2 Hz). Same color code than in *A* and *B*.

DOI: https://doi.org/10.7554/eLife.28935.005

($R_1 \rightarrow V$). In this case, $R_0 + R_1$ and V correspond to the reluctant and the fully-releasable pool, respectively, as observed in our experiments. In the simulations, release was triggered by Gaussian-shaped $Ca^{2+}$ signals with amplitudes of ~22 µM (*Schmidt et al., 2013*). $Ca^{2+}$-dependent recruitment was assumed to be driven by residual $Ca^{2+}$ (*Figure 6B*) with an initial amplitude of 520 nM and a decay constant of 42 ms (*Brenowitz and Regehr, 2007*). During high-frequency stimulation (100 Hz), this residual $Ca^{2+}$ sums linearly during 50 stimuli (*Brenowitz and Regehr, 2007*), building up to a steady state amplitude of ~2 µM. This high concentration of residual $Ca^{2+}$ drove rapid recruitment of SVs from the reluctant pool into the fully-releasable pool causing a transient increase of the latter pool (*Figure 6C*; *Valera et al., 2012*). Acting in concert with slow $Ca^{2+}$ unbinding from the release sensor, which generates a moderate facilitation on the ms timescale (*Bornschein et al., 2013*), SV recruitment resulted in prolonged and facilitated release consistent with our experimental observations (*Figure 6D,E*, compared with *Figure 1*). On the other hand, during low-frequency stimulation (2 Hz), residual $Ca^{2+}$ dropped back to its resting level between pulses and recruited SVs returned to the reluctant pool between stimuli. This resulted in the progressive depletion of fully-releasable SVs, leading to LFD. Renewed driving of the depressed model at high-frequency reproduced a rapid recovery from LFD and was followed by facilitation (*Figure 6D* compared with *Figure 2*), thus illustrating the recruitment of the reluctant pool.

Next, we simulated release using two alternative models. The single pool model incorporates an 'allosteric' release sensor (*Lou et al., 2005*) and a replenishment step at a fixed, $Ca^{2+}$-independent rate (one-pool model, *Wölfel et al., 2007*). The parallel pool model postulates two pools of releasable SVs (SRP and FRP, see introduction) with fast and slow intrinsic release rates (parallel two-pool model, *Wölfel et al., 2007*). In both models, the pools are replenished at a fixed rate independent of $Ca^{2+}$. Both models predict a depression of release during high-frequency stimulation and a strong facilitation during sustained low-frequency stimulation (*Figure 6F,G*). This is in stark contrast to our experimental results that show the opposite behavior of GC synapses. The latter two models successfully described several aspects of release at the depressing calyx of Held synapse that operates with a very large RRP (*Sätzler et al., 2002*) but cannot explain the release mechanisms at GC-PC synapses, which are based on a small number of releasable SVs and a rapid $Ca^{2+}$ driven recruitment of SVs.

Taken together, these findings show that several aspects of our experimental findings, in particular high-frequency facilitation and LFD, can be reproduced by a simple sequential two-pool model with $Ca^{2+}$-dependent recruitment. This does not exclude a contribution of more sophisticated mechanisms like activity-dependent 'a posteriori' modifications (*Wölfel et al., 2007*) or a separate facilitation sensor (*Atluri and Regehr, 1996*; *Jackman et al., 2016*) but hints towards the minimal requirements for sustained release from a small terminal operating with a small number of releasable SVs.

## Recovery from LFD depends on the size of both pools

The two-pool sequential model predicts that the kinetics of refilling of the fully-releasable pool after LFD depends on the state of the reluctant pool. For example, when both the fully-releasable pool and the reluctant pool have been depleted by high-frequency stimulation, the replenishment should

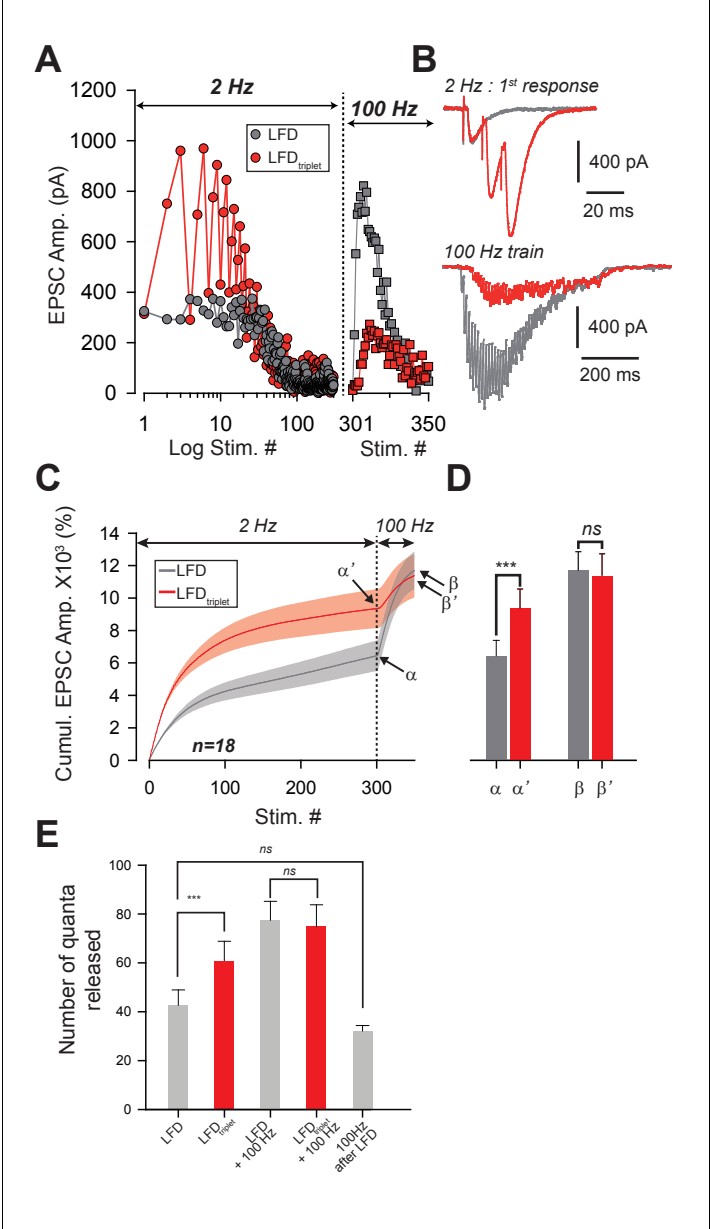

**Figure 4.** Recruitment of reluctant vesicles by high-frequency trains underpins recovery from LFD. (**A**) Superimposition of ESPC amplitudes elicited during LFD (2 Hz, gray circles) or LFD$_{triplet}$ (2 Hz with triplet stimulation at 100 Hz, red circles) in the same PC, and during the recovery from LFD via application of a 100 Hz train (gray squares for a 100 Hz train applied 500 ms after LFD, and red squares for a 100 Hz train applied 500 ms after LFD$_{triplet}$). (**B**) *upper traces*, superimposition of the first EPSCs recorded at stimulus #1 for LFD (gray trace) and LFD$_{triplet}$ (red trace). *Lower traces*, EPSCs recorded during the 100 Hz trains applied 500 ms after LFD (gray trace) or 500 ms after LFD$_{triplet}$. (**C**) Mean values of cumulative EPSC amplitudes during LFD and LFD$_{triplet}$ followed by a recovery train at 100 Hz (n = 18 cells). LFD and LFD$_{triplet}$ were elicited successively in the same PCs. The dashed line indicates the beginning of the 100 Hz trains. α and α' correspond to the cumulative value at the end of LFD protocols, β and β' are values obtained following the recovery trains. (**D**) Mean values of a, a', b andβ' (same y axis as in *C*). (**E**) Estimation of the number of quanta released at GC-PC synapse during LFD, LFD$_{triplet}$ and during the recovery via 100 Hz train (same set of experiments as in *D*). For the panels *D* and *E*, data were compared by using paired *t*-test.

DOI: https://doi.org/10.7554/eLife.28935.006

The following figure supplement is available for figure 4:

*Figure 4 continued on next page*

*Figure 4 continued*

**Figure supplement 1.** The status of the fully-releasable pool does not influence the mobilization of the reluctant pool.
DOI: https://doi.org/10.7554/eLife.28935.007

start first in the reluctant pool, and then the fully-releasable pool can be reloaded. To test this hypothesis, we determined the recovery kinetics of the fully-releasable pool after LFD (*Figure 7A*, red points) and LFD$_{triplet}$ (triplet at 200 Hz; *Figure 7A*, blue points) by applying low-frequency stimulation at 0.033 Hz starting 30 s after the end of the LFD protocol. As shown in *Figure 7A*, full recovery after LFD was described by a single exponential function (tau = 153.8 s, R$^2$ = 0.93), indicating a single-step process. Recovery following LFD$_{triplet}$ also followed a single exponential time course (tau = 212.8 s, *Figure 7A*) but started with a long delay. Averaged time courses show that simple 90 s shift led to an identical monoexponential recovery in both LFD (mean tau = 156.3 s, R$^2$ = 0.98, *n* = 8) and LFD$_{triplet}$ (mean tau = 149.3 s, R$^2$ = 0.96, *n* = 6) (*Figure 7B,C*). The delay preceding the exponential recovery from depression after LFD$_{triplet}$ (~90 s) may reflect the time required to reconstitute the reluctant pool. To test this possibility, the following paradigm was designed: SVs from both the reluctant and fully-releasable pools were depleted by LFD$_{triplet}$ (triplets at 100 Hz), then a 100 Hz test train was applied after a variable resting period (500 ms, 10 s or 1 min of rest after the end of LFD, *Figure 7D*). As shown in *Figure 7D*, amplitudes of EPSCs during the test trains increased proportionally to the length of the resting period. More interestingly, after 1 min of rest, the first responses in the test train were still fully blocked (*Figure 7E*) whereas the amplitudes of the late responses reflecting the recruitment of the releasable pool (after stimuli # 11–12 in the test train) were similar to the corresponding responses in the control train (*Figure 7F*). This demonstrates that even after 1 min of rest, no SVs from the fully-releasable pool were ready to be released, despite partial reconstitution of the reluctant pool.

## Reluctant SVs can be recruited by increasing Ca$^{2+}$ entry

Finally, we investigated whether SVs in the reluctant pool are maintained in a state of low release probability. In this case, the reluctant pool could be affected by modifications of $p_r$. First, we studied the properties of LFD and recovery by 100 Hz train upon an increase in $p_r$. Based on our previous studies, we estimated that an increase of [Ca$^{2+}$]$_e$ from 2.5 mM to 4 mM can increase $p_r$ from 0.25 to 0.67 for SVs in the fully-releasable pool (*Valera et al., 2012*). In the presence of 4 mM [Ca$^{2+}$]$_e$, the number of quanta released during LFD almost doubled as compared to LFD elicited in 2.5 mM [Ca$^{2+}$]$_e$ (*Figure 8A–C*). However, the time course of normalized amplitudes of EPSCs during LFD was barely affected in high [Ca$^{2+}$]$_e$; neither the delay, nor the plateau of LFD or the decay time constant were statistically different in low and high Ca$^{2+}$ conditions (*Figure 8D,E*). This suggests that similar

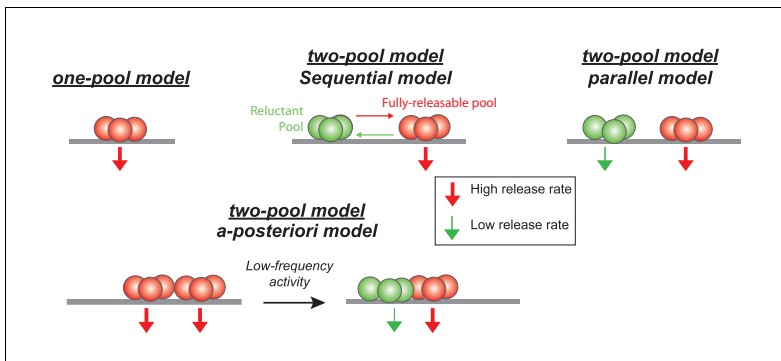

**Figure 5.** Schematics showing different models of SV release at GC-PC synapses. Based on hypothesis proposed in other synapses, presynaptic release could be mediated by a homogeneous pool of release-ready SVs (one-pool model) or through a two-pool model with alternative stages of transition between the two pools. For the two-pool model, releasable SVs are separated in a fully-releasable pool (red SVs) or a reluctant one (green SVs).
DOI: https://doi.org/10.7554/eLife.28935.008

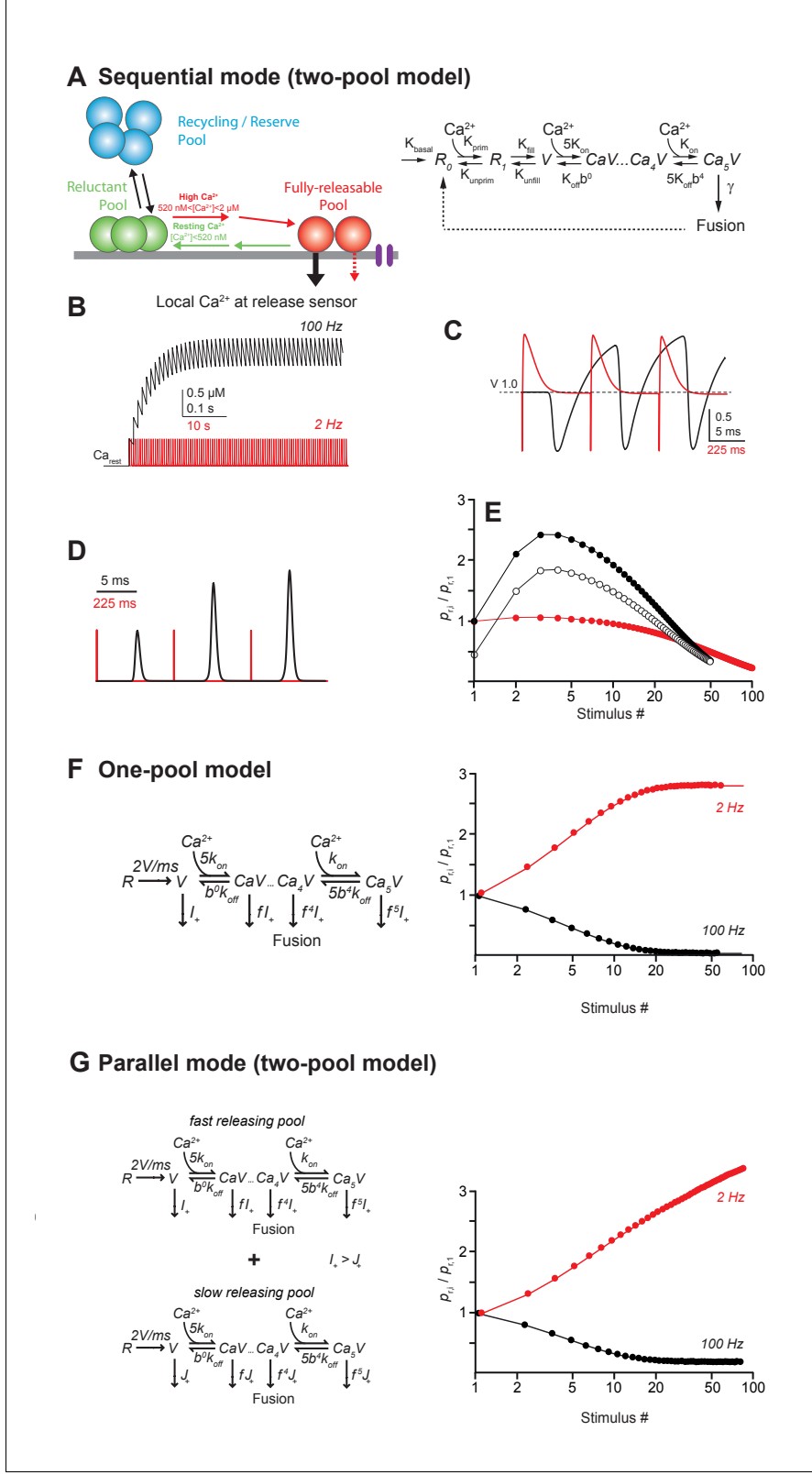

**Figure 6.** Models of LFD and short-term facilitation. (**A**) *Left,* scheme illustrating the sequential model of Ca$^{2+}$ binding and release at the PF-PC synapse. Voltage-dependent calcium channels are represented in purple. *Right,* in the corresponding mathematical model the recycling/reserve pool is contained only implicitly by restoration of the reluctant pool ($R_0+R_1$) from fused SVs (dashed arrow) and via a basal refilling rate ($k_{basal}$). Therefore, this model

*Figure 6 continued on next page*

*Figure 6 continued*

is referred to as sequential 'two-pool' model. During high-frequency stimulation, the residual $Ca^{2+}$ increases, resulting in recruitment of SVs from the reluctant pool into the fully-releasable pool, that is, a temporal increase in the fully-releasable pool that enables short-term facilitation (red arrow). The residual $Ca^{2+}$ generates an additional moderate, short-lasting facilitation due to slow unbinding from the release sensor (dashed red arrow). During low-frequency activation at 2 Hz, the residual $Ca^{2+}$ drops back to resting level between stimuli and SVs recruited to the fully-releasable pool return to the reluctant pool (green arrow). (B) Simulated amplitudes of residual $Ca^{2+}$ during high- (100 Hz, black) or low-frequency (2 Hz, red) activation starting from a resting $Ca^{2+}$ ($Ca_{rest}$) level of 50 nM. (C) Fraction of $Ca^{2+}$ unoccupied SVs in the fully-releasable pool (V) of the release sensor during the initial three activations of a 100 Hz (black) or 2 Hz (red) activation train. Note that during the first three stimuli at low frequency, the fully-releasable pool relaxes to its initial (V = 1, i.e. 100%) from a transient overfilling (V > 1) prior to the next pulse. The fully-releasable pool continues to increase in size during high-frequency activation due to the build-up of residual $Ca^{2+}$ and continuous recruitment of reluctant vesicles (cf. B). (D) Transmitter release rates during three stimuli at high (black) or low frequencies (red), normalized to the first release event (E) Paired-pulse ratios (PPRs) calculated as the ratio of release probabilities in the i-th ($p_{r,i}$) and the first pulse ($p_{r,1}$) during 100 Hz (filled black) or 2 Hz (red) stimulation. Open circles show PPRs during 100 Hz activations started in a previously depressed state. (F) *Left*, one-pool model of $Ca^{2+}$ binding and release according to **Wölfel et al., 2007** consisting of the 'allosteric' sensor model (**Lou et al., 2005**) supplemented with a reloading step of 2 SVs/ms. *Right*, in contrast to our experimental findings, this model generates low-frequency facilitation and high-frequency depression. (G) *Left*, as in F but for two parallel, non-interacting pools of SVs differing in their release rate constants thereby generating a 'fast releasing pool of SVs' (release rate $I_+$ as in F) and a 'slowly releasing pool of SVs' (release rate $J_+ < I_+$ as in F, Wölfel et al., 2007). Both models are restored via $Ca^{2+}$ independent reloading steps of 2 SVs/ms. *Right*, note that similar to the model in F, this simulation generates a high-frequency depression and a low-frequency facilitation. Models in F and G fail to reproduce our experimental findings.

DOI: https://doi.org/10.7554/eLife.28935.009

mechanisms underlying LFD occur at low and high $p_r$ conditions. On the other hand, the recovery from LFD was strongly reduced (maximal recovery 85.5%±11.6 $n$ = 5, **Figure 8D**) with high $[Ca^{2+}]_e$. This indicates that the reluctant pool was partially exhausted after LFD in high $[Ca^{2+}]_e$. Hence, the increase in the number of quanta released during LFD in high $[Ca^{2+}]_e$ probably arises from the recruitment of SVs in reluctant pool.

Next, we tested whether recruitment of the reluctant pool could be affected by impairing the spatiotemporal profile of $[Ca^{2+}]$. EGTA is a slow $Ca^{2+}$ chelator that does not affect $[Ca^{2+}]$ in the nanodomain during single AP but dampens the building-up of $[Ca^{2+}]$ during high-frequency trains (**Schmidt et al., 2013**). Accordingly, LFD and the recovery from LFD by 100 Hz train were monitored before and after application of 10 µM EGTA-AM. Bath application of 10 µM EGTA-AM did not affect the basal release of glutamate (**Figure 8F**, **Schmidt et al., 2013**) and the time course of LFD (**Figure 8F–H**) whereas the recovery from LFD was slowed-down and reduced during the first responses of the 100 Hz train. Statistically significant differences were detected only at stimulus #4 and #5 of the train (p=0.031 for both stimuli, signed rank test, n = 6) (**Figure 8G**). The reduction in the number of reluctant SVs recruited by a 100 Hz recovery train following an alteration of the spatiotemporal profile of $Ca^{2+}$ confirms our simulation that suggests that one transition step from the reluctant state to the fully-releasable is $Ca^{2+}$-dependent.

## Discussion

The present work demonstrates that the release of glutamate and the presynaptic short-term plasticity at GC-PC synapses are shaped by two pools of releasable SVs, namely fully-releasable SVs and reluctant SVs, which are differentially poised for exocytosis. Whether both pools constitute sub-pools of the RRP is still a matter of debate (**Pan and Zucker, 2009**; **Neher, 2015**). Here, we propose that the RRP is restricted to the fully-releasable pool, that is, to the only SVs that can be released by a single AP (see also **Miki et al., 2016**).

### Mechanism underlying the recruitment of reluctant SVs

We cannot exclude that GC boutons are heterogeneous and divided in subsets equipped only with the fully-releasable pool, only with the reluctant pool or with both pools. Nevertheless, we previously

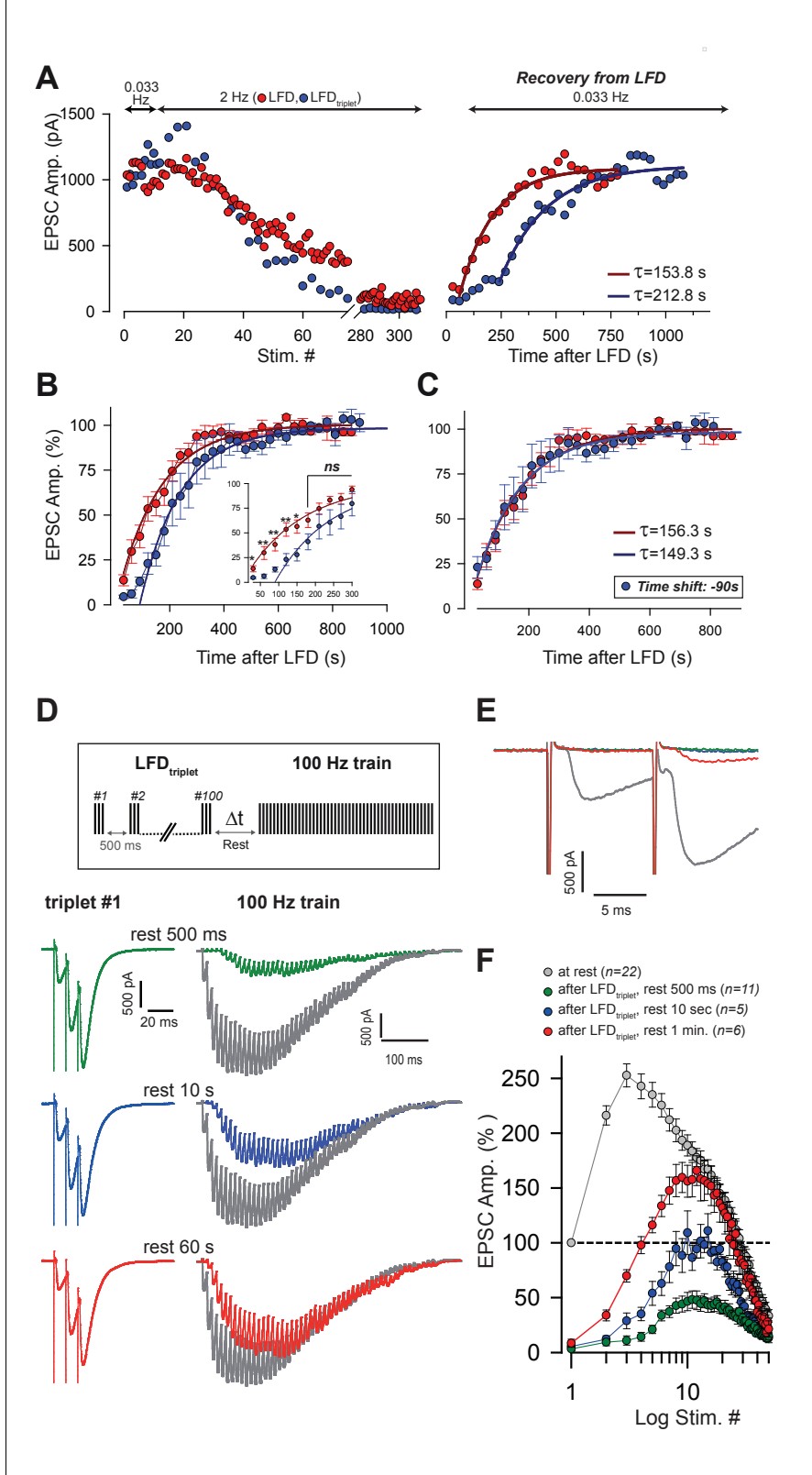

**Figure 7.** Kinetics of recovery from LFD and LFD_triplet. (**A**) *Left*, superimposition of the time course of LFD and LFD_triplet (triplet stimulation at 200 Hz) elicited successively in the same cell after establishing a baseline of at least 10 successive stimuli at 0.033 Hz. *Right*, recovery from depression probed 30 s after the end of both LFD and LFD_triplet by a 0.033 Hz stimulation. For clarity, EPSC amplitudes were plotted against stimulus number in the *left*

*Figure 7 continued on next page*

*Figure 7 continued*

graph and against time in the *right* graph. The thick red and blue lines on the *right* graph show monoexponential fits. Note the delay of recovery after LFD$_{triplet}$, stimulation. (**B**) Normalized mean amplitudes of EPSCs evoked by 0.033 Hz stimulation 30 s after LFD (red points, $n = 8$) and after LFD$_{triplet}$ (triplet stimulation at 200 Hz, blue points, $n = 6$). EPSC amplitudes were normalized to the mean value of EPSCs recorded during a baseline established by stimulation at 0.033 Hz. Thick red and blue lines represent monoexponential fits. (**C**) same values as in *B*, except that the time axis was shifted by 90 s for experimental values obtained during the recovery train applied after LFD$_{triplet}$ (blue points). (**D**) Recovery from LFD$_{triplet}$ probed by 100 Hz as indicated by the stimulation paradigm. *Left*, EPSC traces recorded during the 1$^{st}$ triplet stimulation at 2 Hz. *Right*, EPSC traces recorded during 100 Hz trains applied 500 ms, 10 s or 60 s after LFD protocol ended. Traces are superimposed with EPSCs recorded during a 100 Hz train applied before LFD induction (gray traces). (**E**) Superimposition of the first responses to the trains recorded in *D*. The color code is the same as in *D*. (**F**) Mean values of EPSC amplitudes recorded during 100 Hz trains applied 500 ms, 10 s or 60 s after the LFD protocol ended.

DOI: https://doi.org/10.7554/eLife.28935.010

showed that stimulations of unitary GC-PC synapses and stimulations of beam of PFs gave similar values of the PPR (*Valera et al., 2012*; *Brachtendorf et al., 2015*). Since the ultrafast recruitment of the reluctant pool underlies part of the strong paired-pulse facilitation observed at GC-PC synapses, we believe that most of GC terminals are equipped with both pools. Our simulation and experimental findings show that the recruitment of reluctant SVs is achieved via both Ca$^{2+}$-dependent and Ca$^{2+}$-independent steps. Several presynaptic proteins involved in priming stages bear C2-domains that mediate their actions in Ca$^{2+}$-dependent and Ca$^{2+}$-independent priming processes (*Pinheiro et al., 2016*). Notably, two C2-domain-containing proteins have been implicated in short-term facilitation, synaptotagmin 7 (*Jackman et al., 2016*; *Jackman and Regehr, 2017*) and Munc13s (*Betz et al., 1998*; *Augustin et al., 2001*; *Basu et al., 2007*; *Shin et al., 2010*; *Zhou et al., 2013*). Synaptotagmin 7 is unlikely to play a role in the ultrafast recruitment of SVs from the reluctant pool because of the slow kinetics of its C2A domain (*Hui et al., 2005*; *Brandt et al., 2012* but see *Jackman and Regehr, 2017*). Alternatively, Munc13s can mediate short-term facilitation through the calmodulin-binding domain (*Junge et al., 2004*; *Zikich et al., 2008*) and/or via heterodimerization of the C2A-domain with the Rab3-binding protein RIM (*Camacho et al., 2017*). Interestingly, Munc13s are required in a superpriming step that may be analogous to the transition process described in this study (*Lee et al., 2013*; *Lipstein et al., 2013*; *Ishiyama et al., 2014*). At present, it is not known whether these interactions are fast enough to mediate the transition from the reluctant pool to the fully-releasable one.

In other cerebellar synapses, strong short-term facilitation relies on an ultrafast reloading of release sites (*Saviane and Silver, 2006*; *Miki et al., 2013*). Future experiments will determine whether these fast reloading processes are analogous to the fast recruitment of reluctant SVs at GC to PC synapses. Several scaffolding proteins of the cytomatrix at the active zone such as Bassoon or actin filaments tether SVs in the close vicinity of release sites (*Schoch and Gundelfinger, 2006*; *Siksou et al., 2007*, *2011*; *Gundelfinger et al., 2015*; *Kittel and Heckmann, 2016*) and are required to speed-up the refilling of emptied release sites during high-frequency stimulations (*Hallermann et al., 2010*; *Lee et al., 2012*; *Hallermann and Silver, 2013*; *Miki et al., 2016*). At GC-PC synapses, these filamentous proteins may serve as tracks or molecular motors driving reluctant SVs up to the fusion sites (*Hallermann and Silver, 2013*). Anyhow, both experimental data (*Figure 7*) and simulations (*Figure 6*) suggest that SVs can revert from the fully-releasable status back to the reluctant one, but it remains unclear why the equilibrium shifts toward the reluctant state at low frequencies. The step back to the reluctant state may correspond to an 'a posteriori' mechanism impeding the functioning of active release sites in the low-frequency range (*Wölfel et al., 2007*; *Schneggenburger et al., 2012*). After SV fusion, release sites have to be purged of SV membranes and the time course of this clearance by endocytosis may act as a limiting factor for exocytosis during repetitive activities (*Hosoi et al., 2009*; *Hua et al., 2013*). On the other hand, as proposed for invertebrate synapses (*Silverman-Gavrila et al., 2005*; *Doussau et al., 2010*), LFD may arise from an imbalance in the activation of presynaptic kinases and phosphatases and as such the kinase/phosphatase balance would act as a frequency sensor regulating the equilibrium between the two pools.

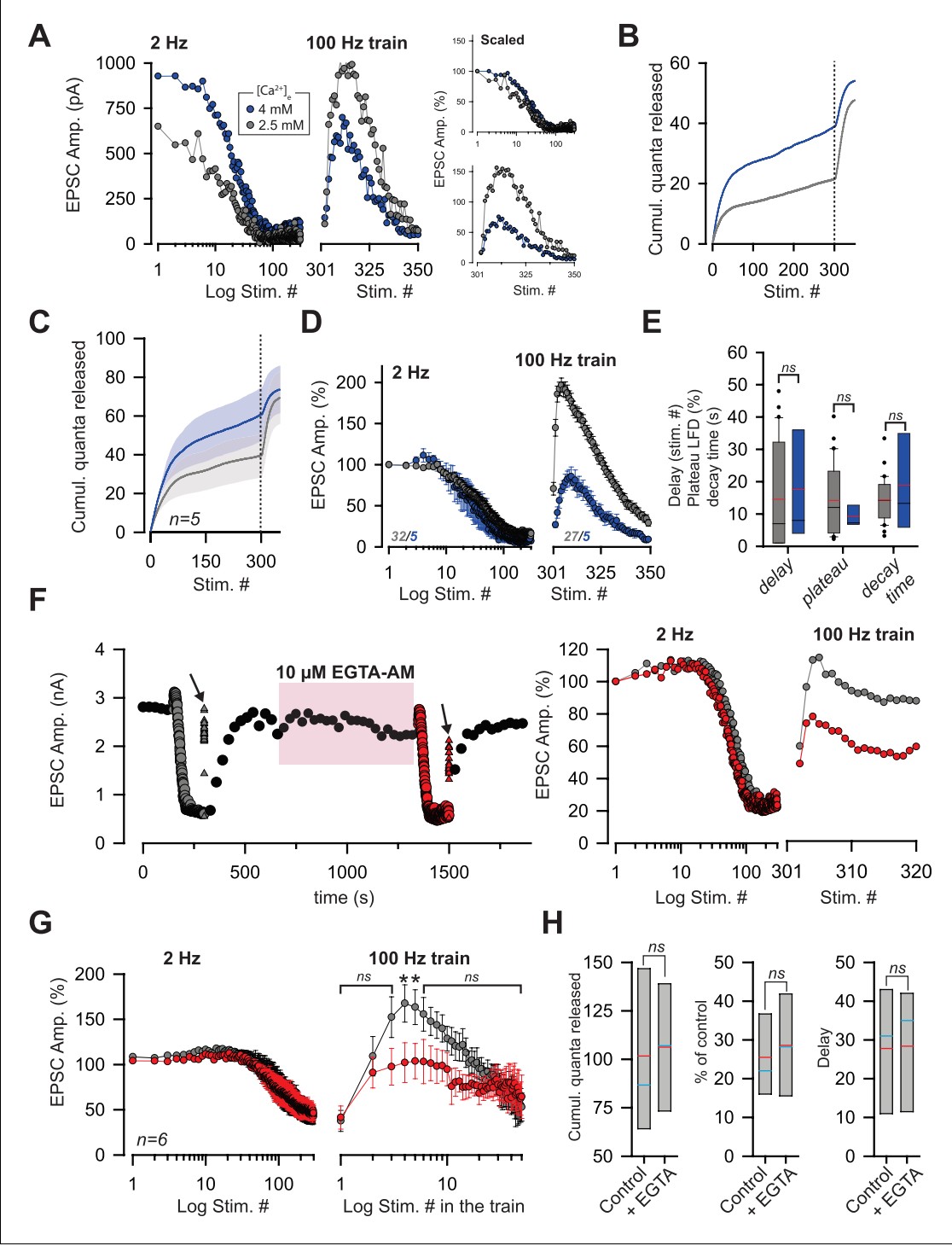

**Figure 8.** The reluctant pool can be recruited by increasing $p_r$. (**A**) Amplitude of ESPCs in the same PC during LFD (*left panel*) and during the recovery from LFD (*middle panel*) by a 100 Hz train before and after an increase in $[Ca^{2+}]_e$ from 2.5 mM (gray points) to 4 mM (blue point). The right panels show the same experiment with EPSC amplitudes normalized with respect to the first response of LFD (**B**) Corresponding cumulative number of quanta release during LFD (stimuli #1 to #300) and during the recovery train (stimuli #301 to #350) in low and high $[Ca^{2+}]_e$. (same color code than in *A*). (**C**), Mean values of the cumulative number of quanta release during LFD (stimuli #1 to #300) and during recovery train at 100 Hz (stimuli #301 to #350) in low and high $[Ca^{2+}]_e$. The dashed line represents the end of the LFD protocol (same color code than in *A*). LFD and recovery trains were recorded in the same PCs. (**D**), Mean values of normalized EPSC amplitudes during LFD (*left panel*) and during the recovery via a

*Figure 8 continued on next page*

*Figure 8 continued*

100 Hz train (*right panel*) in presence of 2.5 mM (gray points, same set of data than in *Figure 1E* for LFD and same set of data than in *Figure 2D* for recovery trains) and 4 mM $[Ca^{2+}]_e$ (blue points). EPSC amplitudes were normalized with respect to the first response of LFD. (E) Box-plots showing the delay (mean number of stimuli before the onset of LFD: 14.5 ± 2.8, *n* = 30 at $[Ca^{2+}]_e$ = 2.5 mM compared to 17.6 ± 8.0, *n* = 5, at $[Ca^{2+}]_e$=4 mM, MWRST, p=0.60), plateau of LFD (mean percentage of initial responses: 14.2 ± 2.0%, *n* = 30 at $[Ca^{2+}]_e$ = 2.5 mM compared to 9.3 ± 1.4%, *n* = 5, at $[Ca^{2+}]_e$=4 mM, p=0.69, MWRST) and decay time (tau: 14.3 s ± 1.3 s, *n* = 30 at $[Ca^{2+}]_e$ = 2.5 mM compared to 18.9 s ± 6.8 s, *n* = 5, at $[Ca^{2+}]_e$=4 mM, p=0.87, MWRST) in presence of 2.5 mM (gray boxes, same set of data than in *Figure 1E* for LFD and same set of data than in *Figure 2D* for recovery trains) and 4 mM $[Ca^{2+}]_e$ (blue boxes). Black and red lines indicate median and mean values, respectively. (F) *left*, typical experiment showing the time course of EPSC amplitudes before and after application of 10 µM EGTA-AM. Black points correspond to EPSCs recorded during 0.033 Hz stimulation. Gray and red symbols correspond to EPSCs recorded during LFD (circles) and during the recovery trains at 100 Hz (triangles) before and after the bath application of 10 µM EGTA-AM respectively. Arrows indicate the application of a recovery train at 100 Hz. The pink box represents the bath application of 10 µM EGTA-AM. *Right*, corresponding ESPC amplitudes normalized with respect to the first response of LFD. (G) Mean EPSC amplitudes recorded during LFD (*left panel*) and during recovery trains (*right panel*) before (gray points) and after (red points) application of EGTA-AM. Note the difference in EPSC amplitudes evoked during the first stimuli at 100 Hz. Numbers at the *bottom* of the graphs indicate the number of cells recorded. (H) Box-plots showing the cumulative number of quanta released during LFD, the plateau of LFD and the delay before the onset of LFD before and after application of 10 µM EGTA-AM (same set of experiment as in *G*). None of these three parameters were statistically different between the two conditions (paired *t*-test, p=0.65, for the cumulative number of quanta released, paired *t*-test, p=0.39 for the plateau of LFD and paired *t*-test, p=0.89, for the delay, *n* = 6). Blue and red lines indicate median and mean values respectively.

DOI: https://doi.org/10.7554/eLife.28935.011

Our calculations indicate that the number of SVs that can be released during high-frequency trains after a full depletion of fully-releasable SVs corresponds to the size of the reluctant pool (*Figure 4*). Since the number of SVs released per bouton (30–40 SVs, *Figures 1* and *4*) during 50/100 Hz trains, LFD, LFD$_{triplet}$ or during recovery trains far exceeds the number of docked SVs counted in one varicosity at GC-PC synapses (4–8 SVs, *Xu-Friedman et al., 2001*), we postulate that the two releasable pools are refilled by other pools (recycling pool, reserve pool) during LFD, LFD$_{triplet}$ and during the late phase of high-frequency trains.

## The segregation of releasable SVs in two pools shapes short-term plasticity

Neuronal networks in the cerebellar cortex have to process sensory information coded at ultra-high frequencies (up to 1 kilohertz, *van Kan et al., 1993*; *Arenz et al., 2008*, *Arenz et al., 2009*). Most of this information is conveyed to the granular layer, the input stage of the cerebellar cortex, via the mossy fiber (MF) pathway. Strikingly, MF-GC synapses can sustain high frequency trains of input by using a specific arrangement of the presynaptic machinery to achieve ultra-fast reloading of SVs (*Saviane and Silver, 2006*; *Rancz et al., 2007*; *Hallermann et al., 2010*; *Ritzau-Jost et al., 2014*). However, it is not clear how these high- frequency inputs are integrated at the GC-PC synapses, the major site for information storage in the cerebellum (*Thach et al., 1992*; *Ito, 2006*; *D'Angelo and De Zeeuw, 2009*). We previously showed that none of the classical mechanisms for facilitation (including the buffer saturation model and residual $Ca^{2+}$, *Pan and Zucker, 2009*; *Regehr, 2012*) involving only an increase in $p_r$ can account for the high paired-pulse facilitation at GC-PC synapses and for its unusual ability to sustain glutamate release at high-frequency trains. During a train of APs at high frequency, the local $[Ca^{2+}]$ at release sites increases (*Schmidt et al., 2013*) leading to an immediate increase in the number of release sites *N* that underlies high values of paired-pulse facilitation (*Valera et al., 2012*; *Brachtendorf et al., 2015*). Here, we show that this increase in *N* arises from the fast recruitment of reluctant SVs into a fully-releasable pool. A fast and sequential recruitment of a 'replacement pool' in 'a docked pool' accounting for paired-pulse facilitation has been recently described at GC-MLI synapses (*Miki et al., 2016*). The GC-PC synapse is unique, because it allows for inactivation of the fully-releasable pool by low-frequency stimulation. The combination of both mechanisms endows these synapses with the striking ability to filter repetitive activities around

2 Hz and to invert the orientation of presynaptic plasticity (full depression versus strong facilitation) depending on the stimulation frequency (*Figure 2*). Finally, during high-frequency inputs, GC-PC synapses are able to set the release glutamate independently of the state of the synapse (synapse at rest or depressed synapse) (*Figure 2*).

### Physiological implications

Our results provide new hypotheses about information processing in the MF-GC-PC pathway. In vivo experiments have shown that in lobules IV and V, GCs responding to joint rotation fire spontaneously at 2–10 Hz due to sustained synaptic inputs from MFs. Upon joint movement, their firing rate shifts instantaneously into burst mode with frequencies ranging from ~50 Hz to 300 Hz (*Jörntell and Ekerot, 2006*). Based on our work, we propose that GC-PC synapses filter GC activity. In the absence of sensory activity, GC inputs to PCs are silenced as most SVs are in the reluctant state. During joint movement, the efficient transmission of high-frequency activity from MF to GC (*Saviane and Silver, 2006*; *Rancz et al., 2007*; *Hallermann et al., 2010*) and from GC to PC (*Valera et al., 2012*) is enabled by the recruitment of the reluctant pool at GC-PC terminals. The efficient filtering of low-frequency activity would enhance the signal to noise ratio and enable PCs to discern sensory signals from spontaneous activity in GCs. Future in vivo studies will reveal whether this model correspond to a realistic process of information in the cerebellar cortex.

## Materials and methods

All experimental protocols are in accordance with European and French guidelines for animal experimentation and have been approved by the Bas-Rhin veterinary office, Strasbourg, France (authorization number A 67–311 to FD).

### Slice preparation

Acute horizontal cerebellar slices were prepared from male C57Bl/6 mice aged 18–25 days. Mice were anesthetized by isoflurane inhalation and decapitated. The cerebellum was dissected out in ice-cold ACSF bubbled with carbogen (95% $O_2$, 5% $CO_2$) and containing (in mM) 120 NaCl, 3 KCl, 26 $NaHCO_3$, 1.25 $NaH_2PO_4$, 2.5 $CaCl_2$, 2 $MgCl_2$, 10 glucose and 0.05 minocyclin. Slices were then prepared (Microm HM650V, Germany) in an ice-cold solution containing (in mM) 93 *N*-Methyl-D-Glucamine, 2.5 KCl, 0.5 $CaCl_2$, 10 $MgSO_4$, 1.2 $NaH_2PO_4$, 30 $NaHCO_3$, 20 HEPES, 3 Na-Pyruvate, 2 Thiourea, 5 Na-ascorbate, 25 D-Glucose and 1 Kynurenic acid (*Millar et al., 2005*; *Wölfel et al., 2007*; *Sakaba, 2008*). Slices 300 µm thick were briefly soaked in a sucrose-based solution at 34°C bubbled with carbogen and containing (in mM) 230 sucrose, 2.5 KCl, 26 $NaHCO_3$, 1.25 $NaH_2PO_4$, 25 glucose, 0.8 $CaCl_2$ and 8 $MgCl_2$ before being maintained in bubbled ASCF medium (see above) at 34°C until further use.

### Electrophysiology

After at least 1 hr of recovery at 34°C, a slice was transferred to a recording chamber. In order to block inhibitory transmission, postsynaptic plasticity and retrograde signaling, $GABA_B$ and endocannabinoid signaling, slices were continuously perfused with bubbled ACSF containing the following $GABA_A$, $GABA_B$, NMDA, CB1 and mGluR1 receptor antagonists: 100 µM picrotoxin, 10 µm CGP52432 (3-[[(3,4-Dichlorophenyl)-methyl]amino]propyl(diethoxymethyl)phosphinic acid), 100 µM D-AP5 (D-(-)−2-Amino-5-phosphonopentanoic acid) and 1 µM AM251 (1-(2,4-Dichlorophenyl)−5-(4-iodophenyl)−4-methyl-N-(piperidin-1-yl)−1H-pyrazole-3-carboxamide) and 2 µM JNJ16259685 ((3,4-Dihydro-2H-pyrano[2,3-b]quinolin-7-yl)-(cis-4-methoxycyclohexyl)-methanone). Recordings were made at 34°C in PCs located in the vermis. PCs were visualized using infrared contrast optics on an Olympus BX51WI upright microscope. Whole-cell patch-clamp recordings were obtained using a Multiclamp 700A amplifier (Molecular Devices, USA). Pipette (2.5–3 MΩresistance) capacitance was cancelled and series resistance ($R_s$) between 5 and 8 mΩ was compensated at 80%. $R_s$ was monitored regularly during the experiment and the recording was stopped when $R_s$ changed significantly (>20%). The membrane potential was held at −60 mV. The intracellular solution for voltage-clamp recording contained (in mM): 140 $CsCH_3SO_3$, 10 Phosphocreatine, 10 HEPES, 5 QX314-Cl, 10 BAPTA, 4 Na-ATP and 0.3 Na-GTP. Parallel fibers were stimulated extracellularly using a monopolar glass electrode filled with ACSF, positioned at least 100 µm away from the PC to ensure a clear

separation between the stimulus artifact and EPSCs. Pulse train and low-frequency stimulation were generated using an Isostim A320 isolated constant current stimulator (World Precision Instruments, UK) controlled by WinWCP freeware (John Dempster, Strathclyde Institute of Pharmacy and Biomedical Sciences, University of Strathclyde, UK). The synaptic currents evoked in PCs were low-pass filtered at 2 KHz and sampled at 20 to 50 KHz (National Instruments, Austin, Texas).

## Simulation of transmitter release and $Ca^{2+}$ dynamics

Models for $Ca^{2+}$-dependent SV fusion and replenishment (*Sakaba, 2008*) were transformed into the corresponding ordinary differential equations and numerically solved using Mathematica 10.0 (Wolfram Research). Release rates were obtained by differentiation of the fused state. Paired-pulse ratios were calculated from the ratios of release probabilities obtained by the integration of release rates. Parameters for the release sensor part of the sequential two-pool model (*Figure 7A,V*, corresponding to fully-releasable SVs) were similar to those used by Sakaba (*Schmidt et al., 2013*) with forward rate $k_{on}$ = 1*$10^8$ $M^{-1}s^{-1}$, backward rate $k_{off}$ = 3000 $s^{-1}$, cooperativity b = 0.25, and release rate $\gamma$ = 5000 $s^{-1}$. Parameters for the replenishment part of this model ($R_0$, $R_1$ representing the reluctant pool) were similar to those given by *Millar et al., 2005* for a phasic synapse and defined empirically. The forward and backward rate constants of $Ca^{2+}$-dependent priming and unpriming were $k_{prim}$ = 8*$10^8$ $M^{-1}s^{-1}$ and $k_{unprim}$ = 120 $s^{-1}$, respectively. $Ca^{2+}$-independent filling and unfilling rates were $k_{fill}$ = 200 $s^{-1}$ and $k_{unfil}$ = 150 $s^{-1}$, respectively. Ninety percent of released SVs were recycled into the unprimed pool, supported by a slow $Ca^{2+}$-independent filling rate of $k_{basal}$ = 0.002 $s^{-1}$, reflecting filling from a reserve pool. Recovery from low-frequency depression (LFD, see results section) was simulated by restarting the model at high frequency with the size of the releasable pool set to the value at the end of the 2 Hz train, and the unprimed pool ($R_0$) fully recovered. The one-pool model (*Figure 7F*) and the parallel two-pool model (*Figure 7G*) and their parameters were taken from *Wölfel et al. (2007)*.

Release-triggering $Ca^{2+}$ signals were simulated as repeated Gaussian curves spaced by interstimulus intervals and adjusted to match the amplitude (22.5 µM) and half-width (5 µs) of the estimated action potential-mediated $Ca^{2+}$ signal at the release sensor of PF synapses (*Schmidt et al., 2013*). In the sequential two-pool model (*Figure 7A*), SV replenishment was driven by residual $Ca^{2+}$ with an amplitude of 520 nM per pulse, dropping exponentially with a time constant of 42 ms and summing linearly depending on the length of the interstimulus interval (*Brenowitz and Regehr, 2007*). For the two other models (*Figure 7G,F*) SV replenishment occurred at a constant rate of 2 SV/ms (*Wölfel et al., 2007*). Resting $Ca^{2+}$ was assumed to be 50 nM.

## Data and statistical analysis

Data were acquired using WinWCP 4.2.x freeware (John Dempster, SIPBS, University of Strathclyde, UK). Analyses were performed using PClamp9 (Molecular Devices, USA), Igor (6.22A) graphing and analysis environment (Wavemetrics, USA). Error bars in figures show SEMs. Student's *t* test or paired *t*-test were performed when data were normally distributed; the Mann-Whitney Rank Sum Test (MWRST) or the signed rank test were used in all other cases. Statistical tests were performed using SigmaPlot 11 (Systat Software). The levels of significance are indicated as *ns* (not significant) when $p > 0.05$, * when $p \leq 0.05$, ** when $p \leq 0.01$ and *** when $p \leq 0.001$.

## Acknowledgements

This work was supported by the Centre National pour la Recherche Scientifique, the Université de Strasbourg, the Agence Nationale pour la Recherche Grant (ANR-15-CE37-0001-01 CeMod) and by the Fondation pour la Recherche Médicale to PI (# DEQ20140329514). We thank the TIGER project funded by INTERREG IV Rhin Supérieur program and European Funds for Regional Development (FEDER, # A31). This work was also supported by a DFG grant to HS (SCHM1838). AMV and KD were funded by a fellowship from the Ministère de la Recherche. AMV was also funded by a fellowship from the Fondation pour la Recherche Médicale. We thank Sophie Reibel-Foisset and the animal facility Chronobiotron (UMS 3415 CNRS and Strasbourg University) for technical assistance. We thank Dr. Frank Pfrieger, Dr. Matilde Cordero-Erausquin and Joanna Lignot (Munro Language Services) for proofreading.

## Additional information

### Funding

| Funder | Grant reference number | Author |
|---|---|---|
| Agence Nationale de la Recherche | ANR-2010-JCJC-1403-1 MicroCer | Philippe Isope |
| Fondation pour la Recherche Médicale | DEQ20140329514 | Philippe Isope |
| Centre National de la Recherche Scientifique | | Philippe Isope |
| Université de Strasbourg | | Philippe Isope |
| INTERREG IV Rhin superieur | FEDER # A31 | Philippe Isope |
| Deutsche Forschungsgemeinschaft | SCHM1838 | Hartmut Schmidt |
| Agence Nationale de la Recherche | ANR15-37-CE37-0001-01 CeModR | Philippe Isope |

The funders had no role in study design, data collection and interpretation, or the decision to submit the work for publication.

### Author contributions

Frédéric Doussau, Conceptualization, Resources, Data curation, Formal analysis, Supervision, Validation, Investigation, Visualization, Methodology, Writing—original draft, Project administration, Writing—review and editing; Hartmut Schmidt, Resources, Software, Formal analysis, Funding acquisition, Investigation, Methodology, Writing—review and editing; Kevin Dorgans, Data curation, Software, Formal analysis, Investigation, Methodology; Antoine M Valera, Software, Formal analysis; Bernard Poulain, Conceptualization, Writing—review and editing; Philippe Isope, Conceptualization, Formal analysis, Supervision, Funding acquisition, Validation, Investigation, Visualization, Methodology, Project administration, Writing—review and editing

### Author ORCIDs

Frédéric Doussau http://orcid.org/0000-0002-3769-1402
Hartmut Schmidt https://orcid.org/0000-0002-9516-423X
Kevin Dorgans http://orcid.org/0000-0003-1724-6384
Antoine M Valera https://orcid.org/0000-0002-0230-9752
Bernard Poulain http://orcid.org/0000-0002-2601-5310
Philippe Isope https://orcid.org/0000-0002-0630-5935

### Ethics

Animal experimentation: All experimental protocols are in accordance with European and French guidelines for animal experimentation and have been approved by the Bas-Rhin veterinary office, Strasbourg, France (authorization number A 67-311 to FD)

### Decision letter and Author response

Decision letter https://doi.org/10.7554/eLife.28935.013
Author response https://doi.org/10.7554/eLife.28935.014

## Additional files

### Supplementary files

• Transparent reporting form
DOI: https://doi.org/10.7554/eLife.28935.012

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
