## [Decision Letter]

Thank you for submitting your article "A frequency-dependent mobilization of heterogeneous pools of synaptic vesicles shapes presynaptic plasticity" for consideration by *eLife*. Your article has been reviewed by three peer reviewers, and the evaluation has been overseen by a Reviewing Editor and Eve Marder as the Senior Editor.

The reviewers have discussed the reviews with one another and the Reviewing Editor has drafted this decision to help you prepare a revised submission.

Summary:

Doussau et al. examined the organization of the synaptic vesicle pool mediating glutamate release and plasticity at the granule cell to Purkinje cell synapse. They found that a long-lasting 2 Hz stimulation depletes a fully-releasable SV pool and that subsequent high frequency stimulation instantaneously recruited a second, reluctant SV pool. This reluctant pool could not be recruited during the 2 Hz single action potential stimulation, yet high frequency triplets applied at 2 Hz recruited both pools. Thus, the authors describe a mechanism that can shift from full depression to maximal strength within a few stimuli at high frequency. This constitutes an efficient mechanism to enhance the signal to noise ratio of distinct sensory inputs.

Although easily releasable and reluctantly releasable SV pools are conceptually not new, the present study beautifully presents a novel and exciting presynaptic mechanism of information processing that could be of general relevance. The activity-dependent recruitment of various pools could act as second layer that can potentially code relevant information about the previous firing history of the cell. This synapse shows remarkable dichotomy in synaptic plasticity, their plastic properties change in an activity-dependent manner. The question of how this happens is intriguing. The authors' suggestion that this phenomenon can be explained by a two-pool vesicle model is solidly built on previous publication, it is logical and carefully assessed in the manuscript. The first part of the manuscript provides detailed experimental data on the parameters of the bidirectional presynaptic plasticity. These experiments are well thought out and carefully executed. The idea of using triplets in Figure 2 is very clever and creative. The quality of the collected data is high. Using modeling, the authors show that the two-pool model is suitable to explain their experimental observations. This section is very thoughtful, the chosen models are a nice representation of the known models, and the author made a good job describing them to a wider audience who might not have modelling experience.

While all the reviewers congratulate the authors on high-quality data and on combination of experiments and modelling, there are some points that should be addressed before publication of the manuscript:

Essential revisions:

At what frequency range can LFD be triggered in this synapse? This might be important because Jörntell and Ekerot show a range of 5-10 Hz for spontaneous activity.

Does the recruitment of the reluctant pool also occur after 25-30 stimuli at 2 Hz, when depression has started to set in but is not yet fully expressed? This might be important when considering the results published by Jörntell and Ekerot. The increase in signal to noise may not be as strong in this case as compared to full depression after 300 stimuli.

How can the authors be sure that they indeed record from a constant and stable set of terminals throughout their experiments? Given a q of 8 pA at a single varicosity Schmidt et al. (2013), the large EPSCs recorded by the authors suggest that in the order of 100 varicosities are typically stimulated, if one PF forms several varicosities per PC and considering multiple AZs (some have two, Xu-Friedman 2001), the number of stimulated PFs could be as low as 20-30. With these assumptions, it is actually a bit surprising that GC stimulation yielded almost the same EPSC amplitudes than PF stimulation. Also, the conclusions with regard to vesicle pools are more complicated because theoretically one group of PFs could have only "fully-releasable" vesicles and another group could have only "reluctant" vesicles. In this case, the overall conclusions would be different, at least when considering a single GC-PC connection rather than an ensemble of such connections. The authors should cover these issues in the Discussion (and rather shorten the discussion of molecular mechanisms).

Figure 1: Experiments involving GC stimulation – how do the authors ascertain that only one GC is stimulated? How do they exclude that the hypothesized change in recruitment of PF does not occur with GC stimulation?

Figure 2: Why does the magnitude of the response differ between 100 Hz after keeping the synapse at rest and 100 Hz after LFD? If the reluctant pool is independent of the fully-releasable pool, they should both have the same amplitude. The model reflects this quite perfectly (Figure 6)!

Subsection “Reluctant SVS can be recruited by increasing Ca^2+^ entry”: "This indicates that.part of the reluctant pool was actually recruited during LFD upon increase in p_r_." Consumption of the reluctant pool during LFD is one interpretation. However, the higher Ca^2+^ could also have a direct effect on recruiting the reluctant pool during the 100 Hz train, assuming that it does not get partially consumed during LFD. If the reluctant pool gets consumed, the total number of quanta released during LFD should be increased when comparing high and low Ca^2+^ conditions. Did the authors observe that?

Subsection “Reluctant SVS can be recruited by increasing Ca^2+^ entry”: This paragraph should end with a conclusion. My take is that buffering Ca^2+^ decreases the recruitment of the reluctant pool. However, increasing external Ca^2+^ also decreased the recruitment of the reluctant pool (see 8A/B). How can this discrepancy be explained?

The authors claim that Synaptotagmin-7 a good candidate for mediating reluctant release. However, due to its slow Ca^2+^ binding kinetics it may not be able to respond within 10-30 ms (see Hui et al. 2005). Please clarify this in the Discussion.

Discussion, subsection “Mechanism of recruitment of reluctant SVs”: "Fully-releasable and reluctant SV differ by their Ca^2+^ sensitivity […]" The evidence for this seems to be somewhat indirect. This could be explicitly discussed or addressed experimentally.

The discussion of molecular mechanisms could be more focused. Any of the proposed mechanisms would have to be differentially implemented for the two pools within the same active zone.

---

## [Author Response]

Essential revisions:At what frequency range can LFD be triggered in this synapse? This might be important because Jörntell and Ekerot show a range of 5-10 Hz for spontaneous activity.

We agree that the frequency range that can trigger LFD is an important issue for cerebellar physiology. Since we hypothesize that LFD may increase the signal to noise ratio, it is important to test whether physiological values of GC spontaneous firing rate (i.e. in absence of sensorimotor inputs) may trigger LFD. in vivo GC recordings was first performed in crus I and IIa by Chadderton et al. (Nature, 2004, PMID 15103377) who found a mean value of 0.4 Hz ± 0.2 Hz. Jörntell and Ekerot (J. Neurosci, 2006, PMID 17093099) found different values and notice that “generally, strong peripheral activation was associated with low spontaneous activity [of GCs] and vice versa”. As mentioned in the Discussion, frequencies chosen in our work for triggering LFD and for the recovery are in the range of the firing frequencies found by Jörntell and Ekerot recorded in GCs responding strongly or moderately to joint movement inputs. However, as mentioned by the reviewers, these GCs fire spontaneously at frequencies higher than 2 Hz (6 Hz ± 3 Hz, Jörntell and Ekerot 2006, Table 3, PMID 17093099).

Accordingly, we performed a new set of experiments in which sustained low-frequencies stimulation at 0.5 Hz, 2 Hz and 5 Hz were successively applied on the same beam of parallel fibers (PFs) and on the same Purkinje cells (PCs). As now shown in the Figure 1—figure supplement 1, LFD was observed at every frequency applied between 0.5 and 5Hz. We also found that LFD can be induced by random frequencies applied in a range of 0.5 Hz to 10 Hz. It should be noted that 5 Hz stimulation led to LFD with similar kinetics than at 2 Hz. These results confirm that LFD can filter a broad range of spontaneous activities.

Does the recruitment of the reluctant pool also occur after 25-30 stimuli at 2 Hz, when depression has started to set in but is not yet fully expressed? This might be important when considering the results published by Jörntell and Ekerot. The increase in signal to noise may not be as strong in this case as compared to full depression after 300 stimuli.

The recruitment of the reluctant pool in function of the status of the synapse (at rest, at the onset of LFD, at the plateau of LFD …) is, indeed, an important issue to understand how information are processed at GC-PC synapses. We therefore designed a specific experiment to address this point.

The strong paired-pulse ratio facilitation during paired-pulse stimulation elicited at high frequencies is one of the most striking particularities of the behavior of the release machinery in GC boutons. As demonstrated in previous publication, an ultrafast increase in *N* (number of release sites) underlies these high PPR values (Valera et al., 2012; Miki et al., 2016). We then considered high PPR values as a signature of the recruitment of the reluctant pool (see also Miki et al., 2016). To test whether the recruitment of the reluctant pool can occur at any time during LFD, we elicited random paired-pulse stimulation at high frequencies during the course of the LFD protocol at 2Hz or 5Hz (see paradigm of stimulation Figure 4—figure supplement 2). As shown in Figure 4—figure supplement 2, extra stimulation leading to paired-pulse high frequency stimulations were associated with unexpectedly high PPR values (that is, PPR>3*z-score of mean PPR value during LFD at 2 Hz and 5 Hz, see Figure 4—figure supplement 2). These high PPR values could be elicited at any time point during the course of LFD protocols and were not correlated to the percentage of inhibition. These new results indicate that the recruitment of the reluctant pool can occur whatever was the state of the fully-releasable pool (available or silenced).

How can the authors be sure that they indeed record from a constant and stable set of terminals throughout their experiments? Given a q of 8 pA at a single varicosity Schmidt et al. (2013), the large EPSCs recorded by the authors suggest that in the order of 100 varicosities are typically stimulated, if one PF forms several varicosities per PC and considering multiple AZs (some have two, Xu-Friedman 2001), the number of stimulated PFs could be as low as 20-30. With these assumptions, it is actually a bit surprising that GC stimulation yielded almost the same EPSC amplitudes than PF stimulation. Also, the conclusions with regard to vesicle pools are more complicated because theoretically one group of PFs could have only "fully-releasable" vesicles and another group could have only "reluctant" vesicles. In this case, the overall conclusions would be different, at least when considering a single GC-PC connection rather than an ensemble of such connections. The authors should cover these issues in the Discussion (and rather shorten the discussion of molecular mechanisms).

We agree that our calculation can only give a rough estimation of the number of quanta released during a given paradigm of stimulation notably because ~10% of PFs establish more than one contact on a single PC (Xu-Friedman et al., 2001, PMID 11517256). Hence, our calculation overestimate the number of PF recruited by external stimulations and therefore underestimate the number of quanta released by varicosity during a given paradigm of stimulation. The percentage of PF establishing more than one active zone per Purkinje is based on observation made on only 39 PFs (Xu-Friedman et al., 2001) and it is not known whether this percentage vary in function of the position of PFs in the molecular layer. Also, 80% of synaptic contact established by PFs on PC are silent (Isope and Barbour, 2002, PMID 12427822) and again, it is not known whether 2 active zones made by a single PF on a same PC are both functional. Anyhow, we added a caveat to mention putative error that can be made in the estimation of the number of quanta released.

GCL stimulation may lead to large EPSC since numerous GC somata are stimulated by extracellular stimulations. This point was not clear in the previous version of the manuscript and we modified Figure 1. It should be noted that similar EPSC amplitudes following molecular layer stimulation (i.e. beam of PFs) and GCL were also observed in other studies (for example, Marcaggi and Attwell, Figure 1, PMID 17110417).

The segregation of GC terminal in groups having only the fully-releasable pool one or only the reluctant one is an interesting hypothesis. This hypothesis is mentioned in this new version of the Discussion (see first paragraph of the Discussion). Nevertheless, we have previously showed that the properties of the PPR (paired-pulse ratio) are identical between pair recordings (that is at unitary GC-PC synapses) or compound stimulations (Valera et al., 2012, PMID 22378898). Since the ultrafast recruitment of the reluctant pool underlies part of the large paired-pulse facilitation observed at GC synapses (Miki et al., 2016, this study), we believe that most of the GC terminals are equipped with both pools.

Figure 1: Experiments involving GC stimulation – how do the authors ascertain that only one GC is stimulated? How do they exclude that the hypothesized change in recruitment of PF does not occur with GC stimulation?

The text mentioning GC stimulation and the corresponding schematic (Figure 1) were probably confusing. Extracellular stimulation in the GC layer led to stimulation of numerous GC somata localized in the vicinity of the tip of the stimulating electrode.

We cannot exclude errors due to changes in GC or PF excitability. The fact that values of PPR are identical between unitary GC-PC synapse (stimulation of a single GC, Valera et al. 2012, PMID 22378898, Schmidt 2013) and at PF-PC synapses (stimulation of beam PF, Valera et al., 2012), suggest that GC and PF excitability are not affected by paired-pulse facilitation. However, one can argue that changes in GC or PF excitability may underlie part of the depression during LFD. Because of differences in membrane properties between the soma and axon (Debanne et al., 2011, PMID 21527732), changes in membrane excitability following repetitive activity should be strongly different in both compartments. The fact that the kinetics of LFD after stimulating GC somata or PFs were not significantly different suggests that, change in excitability, if any, weakly contributes to LFD.

One can also argue that high frequency stimulation may increase PF excitability. However, if an important recruitment of PFs by high frequency trains had taken place through changes in excitability, it would have given rise to a release of glutamate in all conditions and notably after LFD_triplet_. It appears that after LFD_triplet_, the recovery of glutamate release by high-frequency trains can be completely abolished (Figure 3, lower panels, blue points); this suggests that, at best, PF recruitment during these trains could hardly contaminate facilitation induced by presynaptic mechanisms. Hence, a recruitment of PF by an increase in PF excitability cannot underlie paired-pulse plasticity.

Figure 2: Why does the magnitude of the response differ between 100 Hz after keeping the synapse at rest and 100 Hz after LFD? If the reluctant pool is independent of the fully-releasable pool, they should both have the same amplitude. The model reflects this quite perfectly (Figure 6)!

At rest, the first responses at 100 Hz (i.e. the second response of the train) rely both on the fully releasable pool and on the reluctant pool. The high value of facilitation observed at 100 Hz is the consequence of i) the well-described increase in *p_r_* (due to buffer saturation, residual Ca^2+^ …) that leads to a higher recruitment of the fully releasable pool and ii) the ultra-fast recruitment or the reluctant pool. At the end of the LFD protocol, the fully releasable pool has been silenced, then it cannot participate to the first responses of 100 Hz train. In other words, differences in the first responses of 100 Hz train at rest or after LFD are due to differences in the availability of the fully-releasable pool after LFD.

Subsection “Reluctant SVS can be recruited by increasing Ca^2+^ entry”: "This indicates that.part of the reluctant pool was actually recruited during LFD upon increase in pr." Consumption of the reluctant pool during LFD is one interpretation. However, the higher Ca2+ could also have a direct effect on recruiting the reluctant pool during the 100 Hz train, assuming that it does not get partially consumed during LFD. If the reluctant pool gets consumed, the total number of quanta released during LFD should be increased when comparing high and low Ca2+ conditions. Did the authors observe that? Subsection “Reluctant SVS can be recruited by increasing Ca2+ entry”: This paragraph should end with a conclusion. My take is that buffering Ca2+ decreases the recruitment of the reluctant pool. However, increasing external Ca2+ also decreased the recruitment of the reluctant pool (see 8A/B). How can this discrepancy be explained?

In a given presynaptic cterminal, the total number of releasable SVs is supposed to be constant. By increasing extracellular [Ca^2+^]_e_ from 2 mM to 4 mM, we classically observed a strong increase in the basal level of neurotransmitter release (~2 fold increase in EPSC peak amplitudes). Strikingly, this increase in *p_r_* did not affect the properties of LFD when EPSCs were scaled to initial amplitude (onset of depression, plateau of LFD, and decay time constant). Hence, the number of quanta released during LFD upon high [Ca^2+^]_e_ condition was indeed strongly increased; the previous version of Figure 8 was confusing and we modified it in the present version to better illustrate this effect. The strong increase in the number of quanta release during LFD in high [Ca^2+^]_e_ probably reflects a recruitment and a consumption of a substantial fraction of the reluctant pool. Simulations suggest that the fully-releasable SVs can transit back to the reluctant pool during LFD. Potentially, these “newly” reluctant SVs would be recruited by a 100 Hz train but experiments using LFD_triplet_ or high [Ca^2+^]_e_ clearly showed that it was not the case. As mentioned in the Discussion, LFD also involves an “a posteriori” mechanism that might impede the high-frequency dependent recruitment of SVs used during LFD.

Concerning the EGTA experiments, we also believe that Ca^2+^ buffering affects the recruitment of the reluctant pool. Therefore, these experiments suggest that the transition from the reluctant state to the fully-releasable one is Ca^2+^-dependent as suggested by in silico simulation of neurotransmitter release (Figure 6).

The authors claim that Synaptotagmin-7 a good candidate for mediating reluctant release. However, due to its slow Ca^2+^ binding kinetics it may not be able to respond within 10-30 ms (see Hui et al. 2005). Please clarify this in the Discussion.The discussion of molecular mechanisms could be more focused. Any of the proposed mechanisms would have to be differentially implemented for the two pools within the same active zone.

We rewrote the description of the molecular candidates in the Discussion. Whether synaptotagmin 7 (Syt7) participates or not to PPF in a central synapse is indeed a controversial theory. This hypothesis proposed by Regehr and Jackman is based or their recent publication (Jackman et al., 2016, PMID 26738595) in which they showed that rescuing the expression of Syt7 in Syt7 KO mice rescued short-term facilitation in hippocampal and thalamic synapses. This finding is surprising because, as mentioned by the reviewers, slow Ca^2+^ kinetics of Syt 7 is incompatible with the timing of PPF. We added a caveat in the Discussion. We also detailed how Munc13, may be involved in the recruitment of the reluctant pool.

Discussion, subsection “Mechanism of recruitment of reluctant SVs”: "Fully-releasable and reluctant SV differ by their Ca^2+^ sensitivity […]" The evidence for this seems to be somewhat indirect. This could be explicitly discussed or addressed experimentally.

Our model and experiments show that the recruitment of the reluctant pool is Ca^2+^-dependent. We agree that this does not necessarily mean that the two pools differ by their Ca^2+^ sensitivity. Since this hypothesis cannot be drawn from our experiments, we removed it.